# Bioinspired porous three-coordinated single-atom Fe nanozyme with oxidase-like activity for tumor visual identification via glutathione

Da Chen[1], Zhaoming Xia[2], Zhixiong Guo[1], Wangyan Gou[1], Junlong Zhao [3], Xuemei Zhou[4], Xiaohe Tan[1], Wenbin Li[1], Shoujie Zhao[3], Zhimin Tian[1] ✉ & Yongquan Qu [1] ✉

Inspired by structures of natural metalloenzymes, a biomimetic synthetic strategy is developed for scalable synthesis of porous $Fe-N_3$ single atom nanozymes (pFeSAN) using hemoglobin as Fe-source and template. pFeSAN delivers 3.3- and 8791-fold higher oxidase-like activity than $Fe-N_4$ and $Fe_3O_4$ nanozymes. The high catalytic performance is attributed to (1) the suppressed aggregation of atomically dispersed Fe; (2) facilitated mass transfer and maximized exposure of active sites for the created mesopores by thermal removal of hemoglobin (2 ~ 3 nm); and (3) unique electronic configuration of $Fe-N_3$ for the oxygen-to-water oxidation pathway (analogy with natural cytochrome c oxidase). The pFeSAN is successfully demonstrated for the rapid colorimetric detection of glutathione with a low limit of detection (2.4 nM) and wide range (50 nM–1 mM), and further developed as a real-time, facile, rapid (~6 min) and precise visualization analysis methodology of tumors via glutathione level, showing its potentials for diagnostic and clinic applications.

Nanozymes, featured by low cost, high stability, tailorable surface properties, and facile synthesis and storage, deliver enzyme-like activities for prosperous applications in the fields of disease diagnosis and treatments[1–6]. However, nanozymes generally suffer from their lower activity and unsatisfactory specificity in comparison with their natural counterparts, which severely limited their practical applications[7–9]. Fortunately, natural metalloenzymes (cytochrome c oxidase (CcO), superoxide dismutase, etc.) with the well-defined local coordination environments and electronic structures have provided us with ingenious blueprints for the rational design of nanozymes[8,10–12].

Among various nanozymes, the single-atom metal nanozymes with the tailorable chemical, geometric, and electronic configurations of atomically dispersed metals bonding with nitrogen-doped carbon support (M-N-C) possess similar configurations of the active metal centers of natural metalloenzymes, being recognized as alternatives of natural enzymes. Unfortunately, majority of them were synthesized via pyrolysis at high temperatures, leading to structural collapse and part of the buried $MN_x$ units inaccessible to biomolecules[11,13,14]. Also, the strong stacking of those N-doped graphite carbon in nanozymes generally induces the frustrated diffusion of bio-substrates to metal sites. Both mass transfer restriction and active site encapsulation severely decrease the overall activity of those single-atom nanozymes. Besides, pyrolysis can induce the considerable aggregation of single-atom metal due to the carbon loss at high temperatures[15–17].

[1]Key Laboratory of Special Functional and Smart Polymer Materials of Ministry of Industry and Information Technology, School of Chemistry and Chemical Engineering, Northwestern Polytechnical University, 710072 Xi'an, China. [2]Department of Chemistry, Southern University of Science and Technology, 518055 Shenzhen, China. [3]State Key Laboratory of Cancer Biology, Department of Medical Genetics and Developmental Biology, Fourth Military Medical University, 710032 Xi'an, China. [4]Key Laboratory of Carbon Materials of Zhejiang Province, Wenzhou University, 325035 Wenzhou, China. ✉e-mail: zhimintian@nwpu.edu.cn; yongquan@nwpu.edu.cn

Hence, various methods including spatial confinement, defect/vacancy engineering and coordination modulations have been developed to solve those problems[18–20]. However, only part of those inadequacies could be overcome. Thus, seeking for a new synthetic strategy of single-atom metal nanozymes to simultaneously achieve the atomic metal dispersion, modulated electronic structure, elevated mass transport and tailorable coordination environment is still on high demands.

In this work, we report a biomimetic synthetic strategy for the massive and facile preparation of porous single-atom Fe nanozymes (pFeSAN) using hemoglobin (Hb) as Fe-source embedded inside zeolitic imidazolate framework (ZIF-8), which solve the above-mentioned challenges (Fig. 1). Evenly distributed iron atoms in each Hb with a size of 2–3 nm effectively avoid the agglomeration of active sites and creat mesopores in the nanozymes during pyrolysis, thereby maximumly exposing the atomic Fe sites and significantly facilitating the mass transfer of reactants/products in catalysts. Structural characterizations demonstrated the atomically dispersed Fe with a Fe-N$_3$ coordination in pFeSAN. Impressively, pFeSAN delivers high oxidase-like activity, which is 3.3- and 8791- times higher than those of four-coordinated Fe-N single-atom (Fe-N$_4$) and Fe$_3$O$_4$ nanozymes, respectively. Most importantly, the extensive mechanism investigations illustrate that pFeSAN undergo a catalytic pathway of the four-electron reduction of oxygen into H$_2$O, being identical to that of C$c$O[21–24]. Abnormal high concentration of glutathione (GSH) in cells and tissues at millimolar level generally suggests high risk of many diseases[25–27]. Based on the color evolution between colorless 3,3',5,5'-tetramethylbenzidine (TMB) and blue oxTMB enable by oxidase-like pFeSAN and reductive GSH, pFeSAN is developed as a highly selective and sensitive probe for the rapid GSH detection with a low limit of detection (LOD) of 2.4 nM and a wide window (50 nM–1 mM). Then, the pFeSAN-GSH assay is developed for the accurate detection of intratumoral GSH at millimolar levels and as the facile, rapid, and precise visualization methodology of tumor area.

## Results and discussion

### Synthesis and characterization of pFeSAN

We proposed a two-step biomimetic synthesis strategy of pFeSAN by integrating Hb as Fe-resource and ZIF-8 as the precursor of N-doped carbon (Fig. 1). Initially, a mixed methanol solution of Zn(NO$_3$)$_2$, 2-methylimidazole and Hb reacted to produce uniform Hb@ZIF-8 biohybrids with a rhombic dodecahedron shape and an average size of 1081.4 ± 15.4 nm (Supplementary Fig. 1). Compared with the white color of ZIF-8 and brownish red color of Hb, the pale brown color of Hb@ZIF-8 disclosed the successful integration of ZIF-8 and Hb (Supplementary Fig. 2). Consistent with the characterized absorption peak of Hb at ~405 nm, the similar ultraviolet-visible (UV-vis) absorption of Hb@ZIF-8 suggested the integrated Hb in ZIF-8 (Supplementary Fig. 3)[28]. X-ray powder diffraction (XRD) pattern showed that Hb@ZIF-8 inherited the crystallographic structure of ZIF-8, indicating the well-maintained crystal structure (Supplementary Fig. 4). The Infrared C = O bond stretching vibration of amide peak at 1,660 cm$^{-1}$ was observed for both Hb and Hb@ZIF-8, again confirming the successful internalization of Hb with ZIF-8 (Supplementary Fig. 5)[29]. The thermogravimetric analysis suggested a Hb-loading of 40.5 wt.% inside Hb@ZIF-8 (Supplementary Fig. 6). N$_2$ adsorption isothermal analysis displayed an apparently lower surface area of Hb@ZIF-8 (663.8 m$^2$/g) than 850.3 m$^2$/g of ZIF-8 alone, elucidating the partial occupation of ZIF-8 pores by Hb (Supplementary Fig. 7)[29]. Taken together, all characterizations demonstrated the successful immobilization of Hb within ZIF-8.

Subsequently, Hb@ZIF-8 was pyrolyzed at 900 °C under Ar to decompose Hb, remove Zn$^{2+}$ and then give pFeSAN. Compared with Hb@ZIF-8, characterizations of transmission electron microscopy (TEM), scanning electron microscopy (SEM), and high-angle annular dark field-scanning TEM (HAADF-STEM) demonstrated the preserved polyhedral structure of as-synthesized pFeSAN with a reduced size of 685.1 ± 7.3 nm and the formation of porous structure (Fig. 2a and Supplementary Fig. 8). The typical C − N and C = N Infrared vibration peaks instead of C = O stretching vibration of amide were observed at

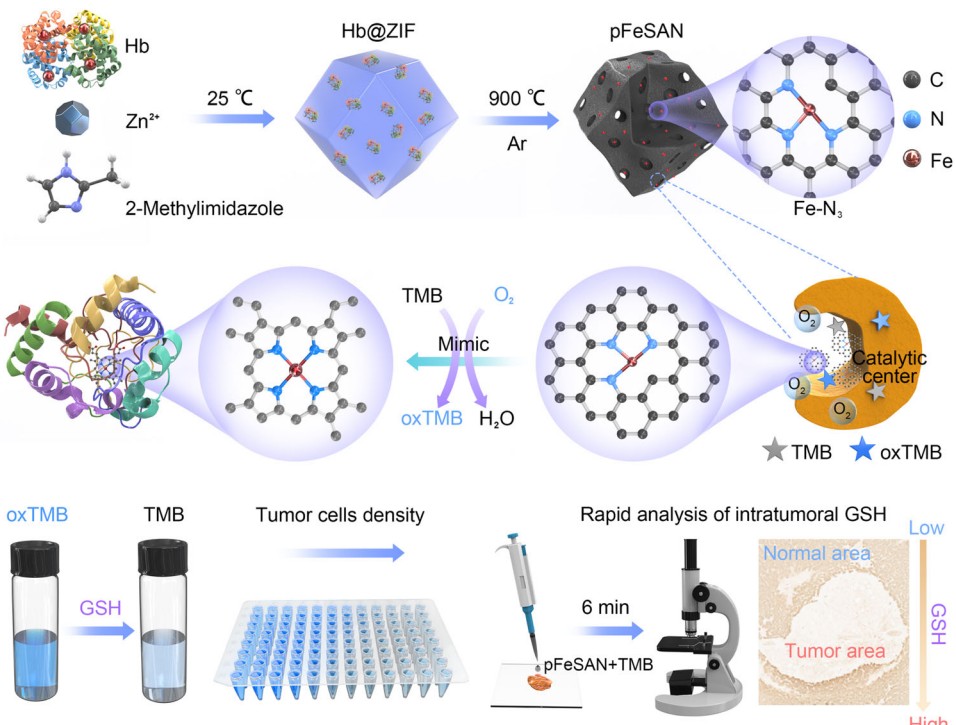

**Fig. 1 | Schematic illustration of the synthesis process and detection effects of pFeSAN.** The pFeSAN was synthesized by a two-step method using Hb@ZIF-8 as precursors, in which mesoporous structures with Fe-N$_3$ sites were formed.

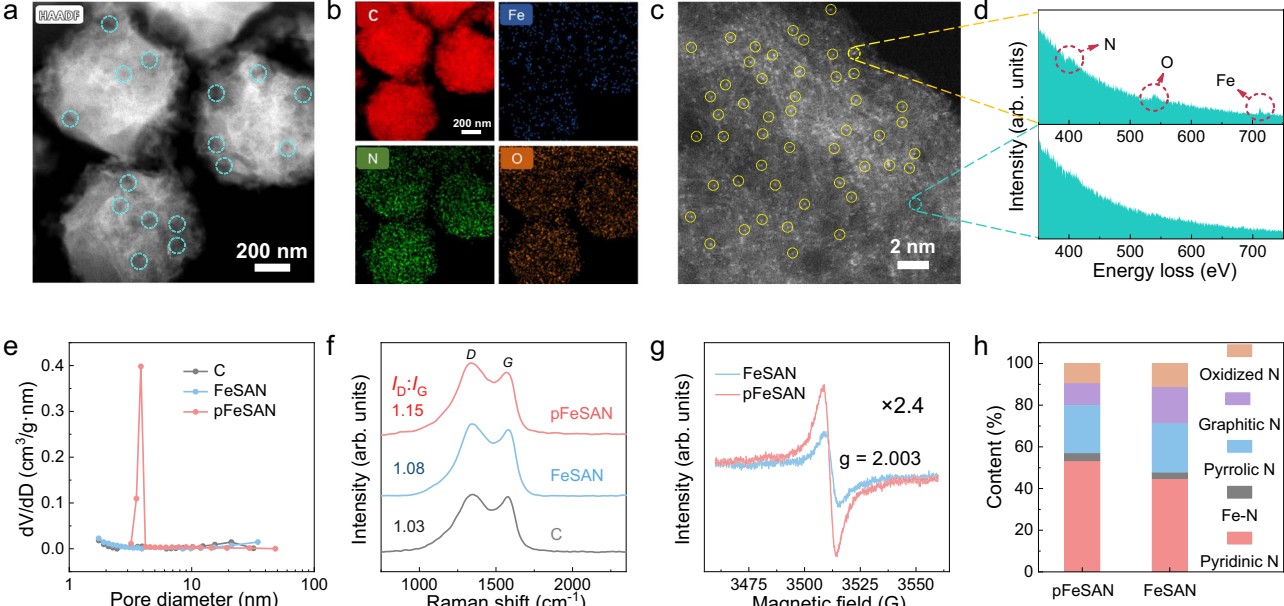

**Fig. 2 | Morphology and structure characterizations. a** HAADF TEM image of pFeSAN. **b** Elemental mapping of C, Fe, N, and O elements of pFeSAN. **c** HAADF-STEM image of pFeSAN showing the atomically dispersed Fe single atom sites as bright dots (yellow cycles marked single atoms). **d** EELS spectra of pFeSAN. **e** Pore size distribution curves of C, FeSAN and pFeSAN. **f** Raman spectra of as-prepared C, FeSAN and pFeSAN. **g** EPR spectrum of FeSAN and pFeSAN. **h** The corresponding percentage of N configurations for FeSAN and pFeSAN. Three times each morphology characterization was repeated independently with similar results. Representative images are shown. Source data are provided as a Source Data file.

1200−1600 cm⁻¹ for pFeSAN, suggesting the thermal decomposition of Hb and the carbonization of Hb@ZIF-8 to give the N-doped carbon (Supplementary Fig. 9)[30]. The XRD pattern of pFeSAN displayed only two broad characteristic diffraction peaks of graphitic carbon at ~24° and ~44° and no peaks related to iron/iron oxides (Supplementary Fig. 10), indicating a high dispersion of iron[11,31]. The energy-dispersive X-ray spectroscopy (EDS) mappings showed the homogeneous distribution of C, Fe, N, and O elements in the entire structure of pFeSAN (Fig. 2b). The aberration-corrected HAADF-STEM with angstrom resolution exhibited many isolated and bright spots (~0.2 nm), demonstrating the presence of the atomically dispersed Fe in pFeSAN (Fig. 2c). As revealed from the electron energy-loss spectroscopy (EELS) of pFeSAN, the point signal at the bright spot (yellow circle) further illustrated the co-existence of the neighboring single Fe and N atoms to give the Fe–N$_x$ configuration (Fig. 2d)[32]. Comparatively, the absence of characteristic peak of Fe at the site of carbon alone (blue circle) again supported the atomic dispersion of Fe. Notably, the cohesive interaction between iron atoms and protein structure of Hb allowed a confinement effect to suppress the iron agglomeration, eventually yielding the atomically dispersed Fe sites.

The commercial Fe₃O₄ nanoparticles, the carbon supports (C) synthesized via the similar process without Hb, the porous carbon supports (pC) obtained using bull serum albumin (BSA) as template and the heme-derived single-atom Fe-N₄ catalysts through pyrolysis (FeSAN) were synthesized as references (Supplementary Figs. 11-16)[33]. A significant number of mesopores (3 - 4 nm) were observed for pFeSAN due to the Hb pyrolysis in ZIF-8 (Fig. 2e), which was also verified by the H₄ type hysteresis loop of the N₂ adsorption/desorption isotherm of pFeSAN (Supplementary Fig. 17a)[31]. Consequently, pFeSAN provided a high surface area of 705.8 m²/g than that of FeSAN (516.6 m²/g) without mesopores (Supplementary Fig. 17b). Raman spectra exhibited a larger $I_D/I_G$ ratio of pFeSAN than those of C and FeSAN, indicating a higher degree of structural defects in pFeSAN (Fig. 2f)[32]. While no characteristic Raman peaks of iron oxides for both pFeSAN and FeSAN further demonstrated the atomically dispersed Fe in two catalysts. Additionally, the electron paramagnetic resonance

(EPR) spectrum of pFeSAN delivered a 2.4-fold higher sharp signal than that of FeSAN at $g$ = 2.003, corresponding to the coordinatively unsaturated iron atoms and abundant N defect sites in pFeSAN (Fig. 2g)[34,35]. Overall, pFeSAN possesses a higher degree of graphitization and abundant nitrogen defect, indicating that the designed biomimetic synthetic strategy enables the effective modulation of the local electronic structure of Fe in pFeSAN and thereby affect their enzyme-like activity.

Afterwards, the chemical states of pFeSAN were investigated by X-ray photoelectron spectroscopy (XPS)[11,31,32]. The N 1$s$ XPS spectra of pFeSAN and FeSAN suggested the existence of five types of N: pyridinic-N (398.3 eV), pyrrole-N (400.5 eV), graphitic-N (401.3 eV), oxidized-N (403.7 eV), and Fe-N peak (399.5 eV, Fig. S18a). Thus, the formation of Fe-N bonds in the N-doped carbon was verified for both catalysts. Compared with FeSAN, pFeSAN possessed a remarkably higher proportion of pyridinic N (Fig. 2h and Supplementary Fig. 18a), which was believed to be important for the oxidase-like activity of single-atom nanozymes[31]. The binding environments of C in pFeSAN and FeSAN can be deconvoluted into three types of C-C/C = C, C-N/O, and O = C-C, indicating the formation of the N-doped graphite-like C (Supplementary Fig. 18b). Compared with the C-N/O peak of FeSAN (285.7 eV), the slight blue shift of the C-N/O peak (286.1 eV) of pFeSAN indicated a stronger electron attraction capability of the Fe atom coordinated with N in pFeSAN than that in FeSAN (Supplementary Fig. 18c)[36]. Thus, the local Fe electronic structure could be effectively modulated by the coordination environments, thereby potentially tailoring the electron transfer and affecting their oxidase-like activity. The O 1$s$ XPS spectra suggested the presence of the absorbed O₂ (530.7 eV), C = O (532.2 eV) and C-OH/C-O-OH groups (533.6 eV, Supplementary Fig. 18d). Especially, the strong and broad peak at 530.7 eV for pFeSAN indicated its strong ability to adsorb and activate oxygen, which could potentially promote the oxidase-like activity[37]. No significant signal of XPS Fe 2$p$ was observed for two catalysts due to the low Fe loadings in pFeSAN (0.36 wt.%) and FeSAN (0.21 wt.%), which were determined by inductively coupled plasma mass spectrometry (Supplementary Fig. 18e).

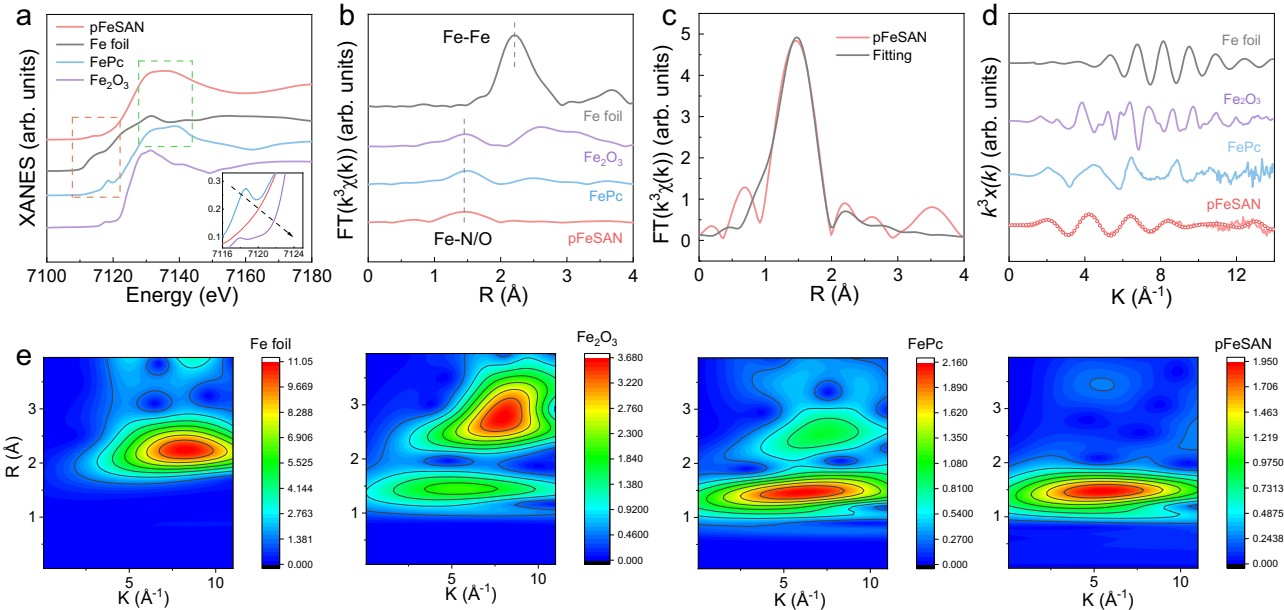

**Fig. 3 | Atomic structure characterization and composition of pFeSAN. a** The Fe K-edge XANES and **b** the Fourier-transformed magnitudes of the experimental Fe K-edge EXAFS spectra of pFeSAN and the reference samples of Fe foil, Fe₂O₃ and FePc, respectively. **c** The experimental FT-EXAFS spectra and fitting curves of pFeSAN. **d** Fe K-edge $k^3$-weighted EXAFS spectra in the $k$ space of Fe foil, Fe₂O₃ and FePc, respectively. **e** Wavelet transform for the Fe K-edge EXAFS signals of pFeSAN and the reference samples of Fe foil, Fe₂O₃ and FePc, respectively. Source data are provided as a Source Data file.

## Atomic structure analysis of pFeSAN

Afterwards, the Fe K-edge X-ray absorption fine structure spectra were employed to probe the local chemical states and coordination environments of pFeSAN with iron phthalocyanine (FePc), Fe₂O₃, and Fe foil as references. The rising-edge position of the X-ray absorption near-edge structure spectra (XANES) of pFeSAN was between those of Fe₂O₃ and Fe foil, and very close to that of FePc, indicating that the atomically dispersed Fe carried a positive charge with the oxidation state slightly over +2 (Fig. 3a). Notably, the pre-edge peak at ~7118 eV, corresponding to the square planar FeN₄ coordination in FePc with $D_{4h}$ symmetry, disappeared in the spectrum of pFeSAN, while the new peak at ~7113 eV revealed the presence of asymmetric iron complexes[38]. Figure 3b showed the Fourier transform (FT) k3-weighted extended X-ray absorption fine structure (EXAFS) spectra. The main peak of pFeSAN at 1.47 Å corresponded to the Fe-N first coordination shell, similar to FePc. The local coordination environment of Fe was resolved by EXAFS fitting (Fig. 3c and Supplementary Table 1), giving a Fe−N distance of 1.91 Å and Fe-N coordination number of 2.8 for pFeSAN. The Fe k-edge $k^3$x(k) curve of pFeSAN showed different oscillation frequencies from those of the reference Fe foil and Fe₂O₃, further indicating that there was obvious Fe−N coordination in the pFeSAN (Fig. 3d). The negligible Fe-Fe signal in 2–3 Å compared with that of Fe foil indicates the atomically dispersed Fe in pFeSAN. This could be further confirmed by the wavelet-transformed EXAFS without peaks corresponding to Fe−Fe bond (2–3 Å in R-space and ~8.3 Å⁻¹ in K-space, Fig. 3e)[34]. The above X-ray absorption analysis demonstrated that the single-atom Fe in pFeSAN existed as the edge-hosted Fe-N₃ moieties. To verify the Fe-N₃ structure in pFeSAN, we then calculated the simulated XANES spectra of Fe-N₃ using the finite difference method of the FDMNES code, which showed good agreement with our experimental data (Supplementary Fig. 19). Previous studies have shown that the stronger adsorption of O₂ on Fe-N₃ than Fe-N₄, indicating the significantly larger oxygen affinity of pFeSAN with the unsaturated Fe-N₃ coordination than that of FeSAN with Fe-N₄ unit[37,39]. Combining the O 1s XPS spectra (Supplementary Fig. 18d), the strong binding between O₂ and pFeSAN can effectively activate oxygen, being a decisive significance for its subsequently enhanced oxidase-like activity[37,39].

Derived from the dynamic light-scattering (DLS) measurements, the average hydrodynamic diameter of pFeSAN in water was 649.7 ± 11.4 nm, consistent with TEM (Supplementary Fig. 20). Moreover, the zeta potentials of pFeSAN in water and acetate buffer (pH 4.0) were −24.3 and 14.2 mV, respectively, indicating the stable dispersion of pFeSAN under various environments (Supplementary Fig. 21). As evaluated, pFeSAN exhibited a long-term stability in water, acetate buffer (pH 4.0) and cell culture medium without the significant changes of the average hydrodynamic diameters during 1-week co-incubation (Supplementary Fig. 22), which benefits the subsequent biomedical applications[40]. Notably, the biomimetic synthesis strategy can be easily scaled up. 13.89 g of pFeSAN was prepared from a single batch reaction of 10L (Supplementary Fig. 23). Importantly, herein, Hb as abundant biomass obtained from stock raising has proved to be a promising renewable source of Fe single atom catalyst. Therefore, this biomimetic synthetic strategy is cost-effective and facile for massive production.

Overall, pFeSAN exhibits multiple advantages including (1) Pyrolysis of natural metalloprotein of Hb (2–3 nm) forms abundant mesopores, leading to the maximum exposure of single metal sites and facilitated mass transfer during catalysis; (2) Benefiting from the protein structure of Hb containing only four Fe atoms, Hb as the Fe-precursor effectively suppresses the Fe aggregation during pyrolysis to give atomically dispersed Fe sites with the maximized utilization efficiency; (3) Unsymmetrically coordinated Fe-N₃ sites deliver a strong adsorption and activation of O₂; (4) High stability under physiological conditions makes pFeSAN suitable for bio-utilizations; (5) Facile, low-cost and scalable synthesis of pFeSAN offers its potential for practical applications. All features suggested the promises of pFeSAN as artificial enzymes for biomedical applications.

## Oxidase-like activity

Oxidase-like activity of pFeSAN was investigated with TMB as substrate in the acetate buffer (pH 4.0, Fig. 4a, b). pFeSAN rapidly catalyzed the oxidation of TMB (1 mM), yielding a blue-colored product with a maximum absorbance at 652 nm. Comparatively, the ZIF-8, Hb@ZIF-8, C, BSA@ZIF, and pC catalysts showed the bare capability for the TMB

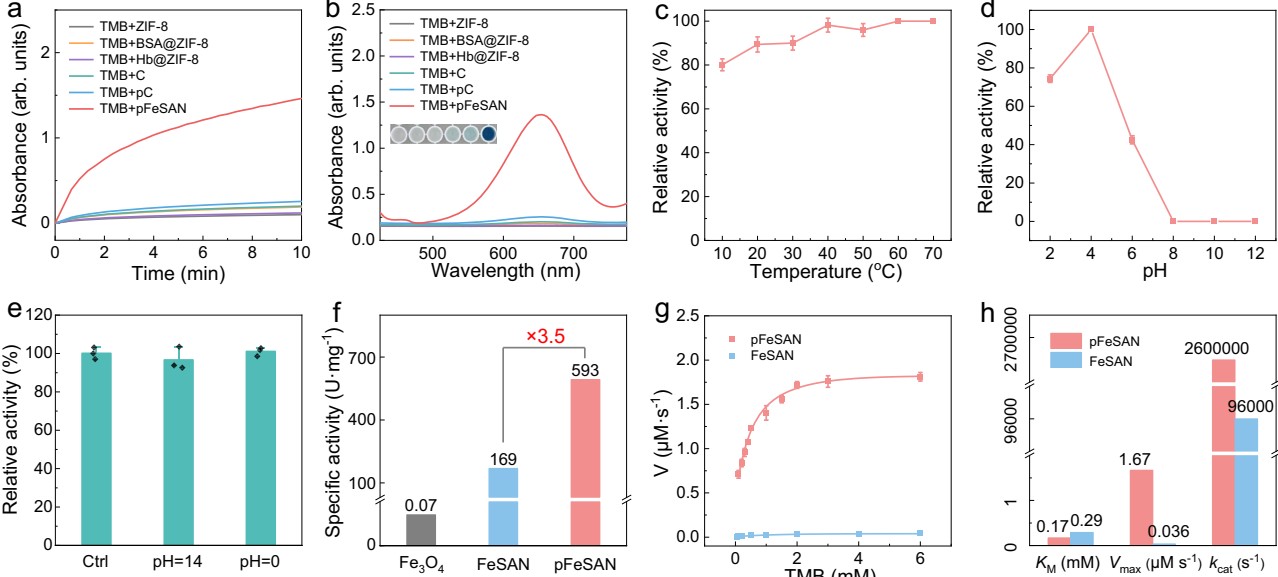

**Fig. 4 | Oxidase-like activity evaluation. a, b** Comparison of oxidase-like activities among various nanozymes. The reaction conditions were 25 °C for 10 min in acetate buffer (pH 4.0). **c, d** Comparison of oxidase-like activities of pFeSAN at various temperature (10–70 °C) and pH (2–12) conditions. **e** The durability of pFeSAN treated with acid or alkali for 24 h. **f** Comparison of the specific activity of oxidase-like catalytic performance of various nanozymes. **g** Steady-state kinetic assay of pFeSAN and FeSAN with TMB as substrate. **h** Comparison of kinetics for pFeSAN and FeSAN. $K_M$ is the Michaelis−Menten constant. $V_{max}$ is the maximal reaction velocity. $k_{cat}$ is the catalytic rate constant. These data are presented as mean values ± SD ($n$ = 3 independent experiments). Source data are provided as a Source Data file.

oxidation, demonstrating the atomically dispersed Fe sites as active centers. Then, the synthetic parameters of pFeSAN were optimized to reach the highest activity. Initially, the pyrolysis temperatures were optimized from 700 to 1000 °C to give the catalysts donated as pFeSAN-T (T represented the pyrolysis temperatures). Their structural characterizations manifested that four pFeSAN catalysts preserved the dodecahedron-like morphology with the abundant mesopores and atomically dispersed Fe species (Fig. 2a–c and Supplementary Figs. 24 and 25). Among various catalysts, pFeSAN-900 delivered the highest catalytic activity (Supplementary Fig. 26), in which the calcination temperature of 900 °C was employed for subsequent investigations. Besides, pFeSAN mentioned hereinafter referred to pFeSAN-900, unless otherwise stated.

Simultaneously, the catalytic activity of pFeSAN was evaluated under various conditions, including environmental pH and temperatures. The catalysts exhibited a highly stabilized oxidase-like activity within a wide temperature window from 10 to 70 °C, suggesting its broad application scenarios under various temperatures (Fig. 4c). Analogy with the natural oxidase enzyme, pFeSAN showed a pH-depended catalytic performance, delivering the highest catalytic activity in weakly acidic media (pH = 4, Fig. 4d)[21,40]. The oxidase-like activity increased with the concentration of pFeSAN and delivered a catalytic activity exceeded 50% at 20 µg/mL (Supplementary Fig. 27). Therefore, the concentration of pFeSAN at 20 µg/mL, room temperature and pH = 4 were chosen as the optimized reaction conditions for future studies. Also, pFeSAN after the pre-treatments with 1.0 M HCl (pH = 0) or 1.0 M NaOH (pH = 14) solutions displayed a satisfactory catalytic stability, which was attributed to the structural robustness of the Fe-N$_3$ conformation (Fig. 4e).

The oxidase-like activity of Fe$_3$O$_4$ and FeSAN as the comparative nanozymes were also evaluated at the same amount of Fe in pFeSAN (Supplementary Fig. 28). Quantitatively, the oxidase-like activities of pFeSNA and other Fe-based nanozymes were evaluated according to the standardized assay protocol[12,41,42]. Compared to FeSAN and Fe$_3$O$_4$, pFeSAN exhibited higher catalytic activity. The specific activity of pFeSAN (593 U mg$^{-1}$) was 3.5 times that of FeSAN nanozyme

(169 U mg$^{-1}$) and 8471 times that of Fe$_3$O$_4$ (0.07 U mg$^{-1}$) (Fig. 4f). Considering the same amount of the atomically dispersed Fe in pFeSAN and FeSAN, their dramatical difference in the oxidase-like activity could be only attributed to the well-regulated coordination environments and the constructed mesopores of pFeSAN. Then, the steady-state kinetic analysis was performed on FeSAN and pFeSAN. The typical Michaelis−Menten curves were obtained by plotting the corresponding initial reaction rates and substrate concentrations (Fig. 4g). The derived $K_M$ (Michaelis constant), maximum reaction velocity ($V_{max}$), and $k_{cat}$ (catalytic constant), were used to assess the binding affinity between TMB substrate and nanozymes and the oxdiase-like activity of nanozymes, respectively (Supplementary Fig. 29 and Table 2)[21,32]. pFeSAN delivered a lower $K_M$ value (0.17 mmol/L) than FeSAN (0.29 mmol/L), suggesting a better affinity between TMB and pFeSAN. Meanwhile, the values of the $V_{max}$ and $k_{cat}$ of pFeSAN were both higher than those of FeSAN. $V_{max}$ of pFeSAN (1.67 µM/s) was 46.4-fold higher than that of FeSAN (0.036 µM/s). $k_{cat}$ of pFeSAN was 2.6×10$^6$/s, which was 27.1-fold higher than that of FeSAN (9.6 × 10$^4$/s, Fig. 4h). Consequently, the catalytic efficiency ($k_{cat}/K_M$) of pFeSAN (1.53 × 10$^7$ mM$^{-1}$s$^{-1}$) was 46.2-fold higher than that of Fe-N$_4$ (3.31×10$^5$ mM$^{-1}$s$^{-1}$). Overall, all these kinetic parameters including $K_M$, $V_{max}$, $k_{cat}$ and $k_{cat}/K_M$ of pFeSAN showed the highest catalytic performance among various catalysts, indicating a distinctly positive contribution of the mesoporous structure (3 - 4 nm) and large surface area to the oxidase-like activity of the pFeSAN (Fig. 2e and Supplementary Fig. 17b). Also, the mesoporous characteristics of pFeSAN delivered a 34.6% higher TMB adsorption capacity of pFeSAN than that of FeSAN in aqueous solutions (Supplementary Fig. 30). Besides, pFeSAN exhibited 5,882 and 1,176 times higher oxidase-like activity than those of the previously reported nanozymes of Pt and Mn$_3$O$_4$, respectively (Supplementary Fig. 31)[43,44]. All results demonstrated that pFeSAN possessed a higher affinity towards TMB substrate and delivered a higher intrinsic catalytic activity than FeSAN, which could be attributed to (1) the modulated local electronic structures for a strong substrate affinity and (2) the mesoporous structure of pFeSAN for the facilitated mass transfer and accessibility of Fe sites.

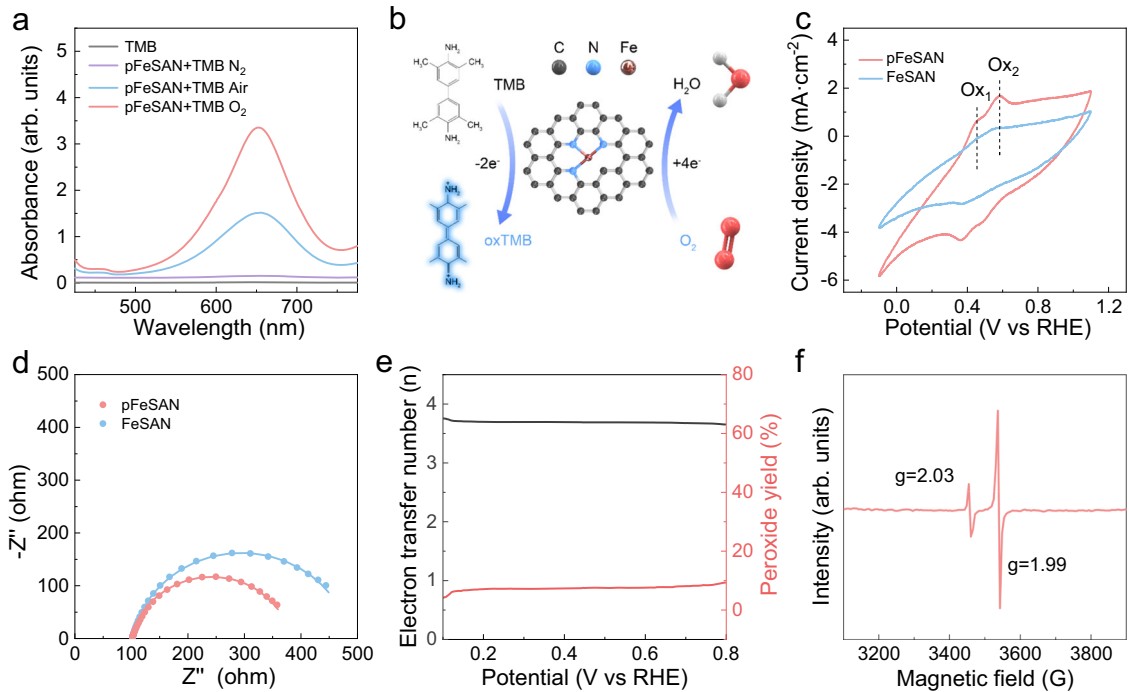

**Fig. 5 | Oxidase-like catalytic mechanism of pFeSAN. a** UV-vis absorption spectra of pFeSAN + TMB in air, $O_2$-saturated, and $N_2$-saturated acetate buffer (pH 4.0). **b** Schematic illustration of oxidase-like characteristics of pFeSAN-catalyzed TMB oxidation. Shadings of TMB represent the oxidized blue TMB. **c** CV curves of pFeSAN and FeSAN in pH 4.0 acetate buffer containing TMB. **d** EIS spectra of pFeSAN and FeSAN. **e** Calculated electron transfer number derived from rotating ring-disk electrode and $H_2O_2$ yields of the pFeSAN. **f** EPR spectra of pFeSAN in the presence of excess phenyloxoiodine at 77 K. Source data are provided as a Source Data file.

## Catalytic mechanism

Next, the catalytic mechanism of pFeSAN was explored. Initially, the $O_2$ levels were controlled for the catalytic TMB oxidation to identify the roles of oxygen[21,40]. pFeSAN showed negligible oxidase-like activity in the nitrogen-degassed environments, indicating that pFeSAN did not serve directly as the oxidants (Fig. 5a). Comparatively, pFeSAN delivered 2.2-fold higher oxidase-like activity in the $O_2$-saturated solution than that under the air-saturated solution, illustrating $O_2$ indeed as the oxidants. Analogous with the majority of the previously reported oxidase-like nanozymes, pFeSAN as oxidase mimetics with oxygen as electron acceptor might generate reactive oxygen species (ROS, e.g.·OH, $^1O_2$,·$O_2^-$) to oxidize TMB[1-3]. To examine this point, the ·$O_2^-$ inhibitor superoxide dismutase, ·OH scavenger mannitol, and $^1O_2$ quencher furfuryl alcohol were introduced into the catalytic reactions[45,46]. Surprisingly, the oxidase-like activity of pFeSAN was only slightly reduced in the presence of those quenchers, suggesting the barely generated free ROS herein (Supplementary Fig. 32a). Furthermore, the trapping agent 5,5-dimethyl-1-pyrroline *N*-oxide was also employed to probe the possible involved active species by electron paramagnetic resonance (EPR)[23,45]. No apparently detectable signals of any ROS further proved that ROSs were not the main intermediates for the oxidase-like activity of pFeSAN (Supplementary Fig. 32b), which was very different from the majority of the previously reported nanozymes[1-3,40]. These results imply that pFeSAN may mediate the complete $O_2$-to-water reduction without the release of free ROS during oxidation, analogy with the catalytic pathways of natural C*c*O (Fig. 5b)[23,45]. Hence, we speculated that the oxygen activation process of pFeSAN might be similar to that of natural C*c*O, in which the Fe(IV) = O was considered to be the key intermediate in catalytic cycles of pFeSAN[47].

Afterwards, the electron transfer process analyzed by electrochemical methods was explored to understand the oxidase-like mechanisms of pFeSAN. Two pairs of oxidation-reduction peaks were observed in the cyclic voltammetry curve of TMB in the acetate buffer (pH=4.0, Fig. 5c)[40,48]. These profiles indicated that the TMB electro-oxidation catalyzed by pFeSAN and FeSAN proceeded via a two successive one-electron oxidation: (1) to yield an intermediate product TMB-free radical and then (2) to give the completely oxidized product quinonediimine. Compared with FeSAN, the redox peak intensity of TMB in the presence of pFeSAN was much stronger, indicating more available active sites of pFeSAN for more efficient TMB oxidation. The electrochemical impedance spectroscopy (EIS) measurements were conducted to reveal the charge-transfer kinetics between TMB and active sites. The semicircle diameter of pFeSAN was smaller than that of FeSAN, indicating a lower interfacial resistance (407.3 to 298.6 Ω) and higher charge transfer efficiency of pFeSAN (Fig. 5d)[49,50].

To explore the electron transfer path during the oxidation, the rotating ring-disk electrode (RRDE) tests were performed. Figure 5e showed that the $H_2O_2$ yield of pFeSAN remained below 7.5% over a wide potential range of 0.1–0.8 V. Derived from the RRDE test, the average electron transfer number (n) of pFeSAN was 3.7, indicating the oxygen activation on the atomically dispersed Fe through a four-electron oxygen reduction reaction pathway[20,22,45,51]. This process requires four $H^+$ and four electrons ($O_2 + 4H^+ + 4e^- \rightarrow 2H_2O$) for the complete $O_2$-to-$H_2O$ reduction. Thus, the electrochemical understandings of these stepwise proton and electron transfers reveal the essence of pFeSAN for its oxidase-like performance.

These results indicated that pFeSAN mediated the complete $O_2$-to-$H_2O$ reduction without the ROS generation, consistent with EPR results (Supplementary Fig. 32b). Previous studies demonstrated that the reactive intermediate of Fe(IV) = O was very important for the catalytic oxidative reactions of natural C*c*O[47,52]. As the common oxidant, the intermediate of Fe(IV) = O, which usually presents in the catalytic cycle of natural oxidases, is considered as the active transient state. To verify the presence of the Fe(IV) = O intermediate of pFeSAN, the EPR spectrum of the pFeSAN-enabled oxidation with excessive

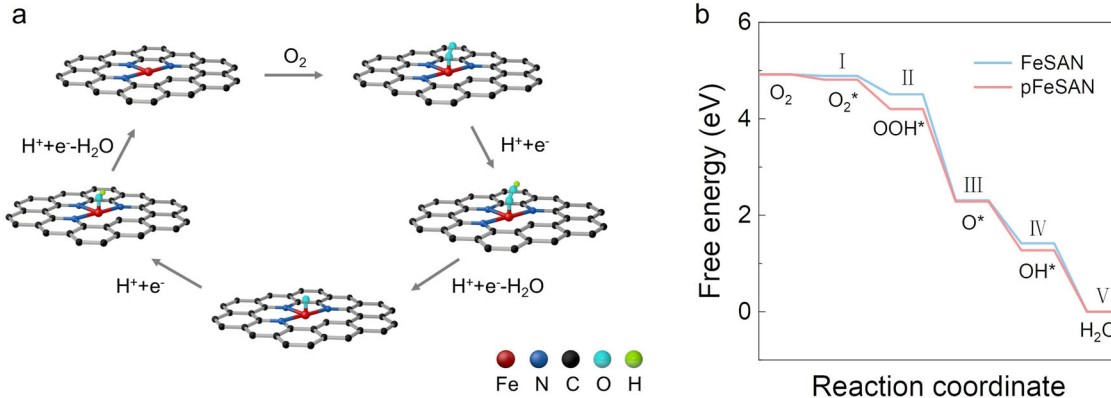

**Fig. 6 | DFT theoretical calculation of oxidase-like activity over pFeSAN. a** Proposed reaction pathways of $O_2$ reduction to $H_2O$ with optimized adsorption configurations on pFeSAN. **b** Corresponding free energy diagram for oxidase-like reaction on FeSAN and pFeSAN. Source data are provided as a Source Data file.

phenyloxoiodine was recorded at 77 K. A typical diamond-shaped sign signal at $g = 2.03$, consistent with $\eta^2$-peroxo heme species, indicated the formation of $Fe(IV) = O$ intermediate in pFeSAN for oxidation (Fig. 5f)[21,23]. Therefore, the oxidase-like activity of pFeSAN proceeds through the $O_2$-to-$H_2O$ pathway, analogy with that of natural C*c*O.

To further reveal the origins of the oxidase-like activity of pFeSAN, density functional theory (DFT) calculations were performed on both Fe-$N_3$ and Fe-$N_4$ units. According to the proposed catalytic pathway of pFeSAN (Fig. 6a), the models of the isolated Fe sites with three- or four-coordinated pyridinic N incorporated within graphene matrix were built for calculations (Fig. 6b)[21,23]. In acidic media, an $O_2$ molecule underwent through (I) the initial adsorption on the isolated Fe sites to give *$O_2$; (II) the subsequent protonation to form *OOH on the top of Fe atoms; (III) the dissociation of the *OOH intermediate associates with $H^+$ to give the active *O intermediates and $H_2O$; (IV) the protonation of *O to give *OH; and (V) the recombination of *OH and $H^+$ to generate $H_2O$. The calculated adsorption energy of *$O_2$ on Fe-$N_3$ was 0.11 eV, which was larger than that of *$O_2$ on Fe-$N_4$ (0.03 eV), indicating a stronger $O_2$ adsorption on pFeSAN and thereafter its effective activation with the weakened O-O bond strength and elongated O-O bond length (1.297 Å) in comparison with 1.23 Å of free $O_2$ and 1.295 Å of *$O_2$ on Fe-$N_4$. DFT calculations indicate the step III (*OOH + $H^+ + e^- \rightarrow$ *O + $H_2O$) as the rate-determined step. In comparison with that of FeSAN (2.20 eV), the smaller $\Delta G_{III}$ of pFeSAN (1.92 eV) suggests its higher oxidase-like activity. Thus, the coordination-unsaturated Fe in Fe-$N_3$ exhibits the strong adsorption and activation of $O_2$, promotes the O-O cleavage of the adsorbed *OOH and generates the active *O intermediates, thereafter greatly improving the oxidase-like activity of pFeSAN.

## GSH detection

GSH is an essential antioxidant to maintain the redox balance of biological systems, and its abnormal levels are commonly associated with various diseases including tumors, infections, and neurodegenerative disorders[25–27,53–57]. The current analytical techniques often require sophisticated equipments and time-consuming sample preparation processes[25,53]. Especially, the GSH levels of tumor cells (0.5–1.0 mM) are much higher than those normal cells. However, its lack of a facile yet accurate method to detect GSH at high concentrations in tumor tissue. GSH with the antioxidant capability can reduce blue oxTMB into colorless TMB, thus providing a quantitative methodology to probe the concentration of GSH and offering a visual methodology to monitor the GSH distribution in biological tissues (Supplementary Fig. 33). With the pre-mixture with GSH, the absorbance of the TMB-pFeSAN system at 652 nm hardly increased with time due to the reductive ability of GSH (Supplementary Fig. 34). Besides, the oxidase-like activity and FT-IR spectrum of the GSH-

pretreated pFeSAN were analogous to that of the untreated pFeSAN, confirming that GSH did not affect the oxidase-like activity of pFeSAN but only reduced the blue ox-TMB into colorless TMB (Supplementary Figs. 35 and 36).

Inspired by the above results, the combination of the reductive GSH and the oxidase-like pFeSAN establishes a simply and rapidly colorimetric GSH assay system (Fig. 7a). As expected, the absorbance decreased with the increased concentrations of GSH and exhibited a linear relation within a broad GSH window (0.05 µM–1.0 mM, Fig. 7b and Supplementary Fig. 37). The detection systems were featured by a broad range of GSH and a low derived LOD of 2.4 nM, which were superior to the majority of other previously reported GSH detection systems (Fig. 7c, Supplementary Fig. 37, and Table 3)[26,55]. Moreover, the color changes of the corresponding measurements in the presence of GSH with various concentrations could be clearly distinguished by the visual inspection (Fig. 7b insert), giving the prerequisites for subsequent visualization of the tissue GSH assay. Furthermore, the pFeSAN-based GSH detection system could resist the interferences of various metal ions, amino acids, and proteins, validating well anti-interference and specificity of the pFeSAN-based GSH detection system (Fig. 7d). The recovery tests were also employed to evaluate the accuracy of this method by standard addition method. The recovery rates of the pFeSAN-based GSH detection system were in the range of 98–106.1% in compassion with the recovery rates of commercial GSH assay kit in the range of 85–107.5%, demonstrating the satisfactory feasibility in practical sample analysis (Supplementary Table 4). Besides, all relative standard deviations (RSD) were <2.5%, which illustrated the accuracy and practically of the pFeSAN-based bioanalysis system for the GSH detection, especially for those at high GSH levels.

The identification and analysis of tumor is particularly important for the survival of cancer patients. Therefore, it is highly desirable to develop sensitive, specific, and rapid methods to diagnose tumor. As a cancer biomarker, the high GSH levels (0.5–1.0 mM) play important roles in maintaining intracellular redox homeostasis in tumor cells. Fortunately, the exceptionally high detection upper limit of pFeSAN is well suitable to detect GSH with high levels. pFeSAN was applied to monitor the GSH levels in normal liver cells (AML12) and liver tumor cells (Hepa 1–6). The absorption of AML12 cells and Hepa 1–6 cells at 652 nm gradually declined with the increased number of cells from 1000 to $8 \times 10^5$ due to the effective reduction by the intracellular GSH. Comparatively, the inhibition efficiency of two types of cells on the chromogenic reaction was very different, which was apparently distinguished by the naked eye (Fig. 7e and Supplementary Fig. 38). The GSH level of Hepa 1–6 cells was higher than that of normal AML12 cells at the same cell density. When the cell densities reached $0.5 \times 10^5$/mL, the absorbance of the two colorimetric systems was significantly

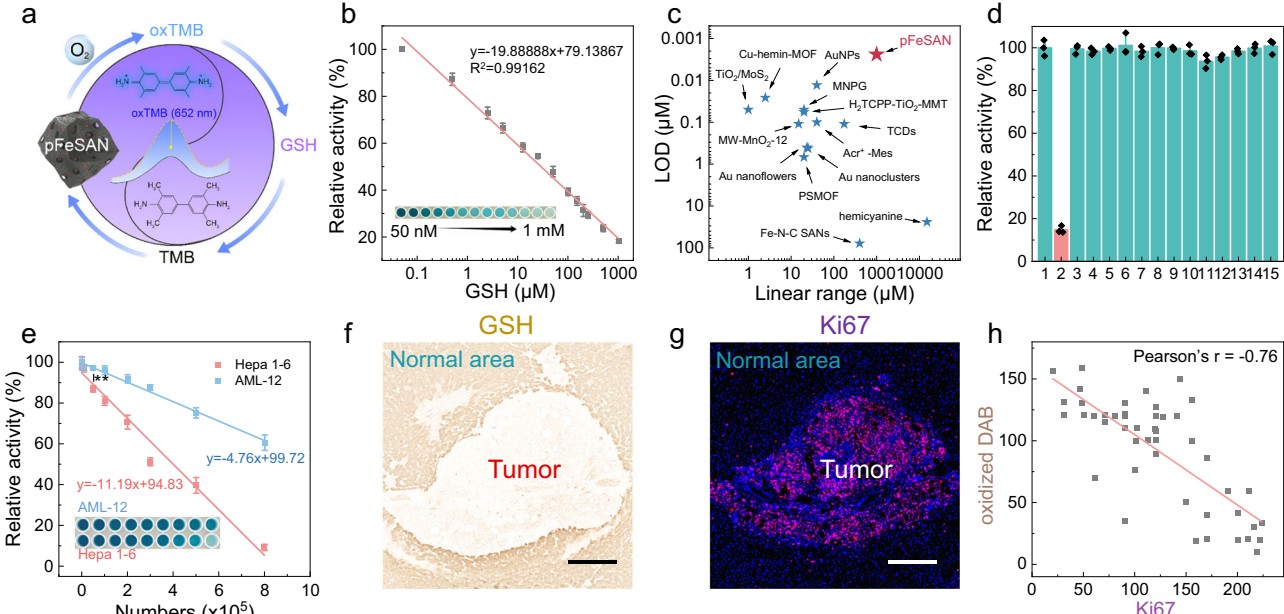

**Fig. 7 | Analytical performance of the pFeSAN-based colorimetric system for GSH detection. a** Schematic illustration of the pFeSAN-based GSH biosensing system. **b** The linear relationship between the relative activity and the GSH concentration ranges from 50 nM to 1 mM. Inset shows the photograph of reaction solution in the different concentration of GSH. **c** Comparison of the performance of the detection of GSH with different methods. **d** Selectivity toward GSH of the pFeSAN-based colorimetric system against common interfering chemicals. The numbers from 1 to 15 are Ctrl, GSH, Na⁺, Ca²⁺, Mg²⁺, K⁺, glutamic acid, tryptophan, arginine, glycine, cysteine, ascorbic acid, glucose, glucose oxidase and BSA. **e** The linear relationship between the variation of relative activity and the number of cells. The inset: photographs of reaction solution with different cell numbers.

$**p = 0.0035$. **f** DAB staining was performed to evaluate the level of GSH in tumor-bearing liver tissue. Scale bar = 200 μm. A representative image of three replicates from each group is shown. **g** Ki67 staining was performed to assess the proliferation of tumor cells in tumor-bearing liver tissue. Scale bar = 200 μm. A representative image of three replicates from each group is shown. **h** Pearson's correlation coefficient of the oxidized DAB (brown)/Ki67 (red fluorescence), which appeared in tissue slices of tumor-bearing liver tissue (Pearson's coefficient $r = -0.76$). These data are presented as mean values ± SD ($n = 3$ independent experiments). Statistical significance was determined by two-tailed $t$ test ($**p < 0.01$). Source data are provided as a Source Data file.

different ($**p < 0.01$). The above results suggested that pFeSAN-based colorimetric systems could serve as a promising platform for the rapid and ultrasensitive GSH detection.

Analysis of the tumor area in clinical practice enables the identifications of the boundary between tumor and normal tissue and the site of surgical resection[58]. The tumor microenvironment usually expresses an excess of GSH in comparison with the health counterparts. Herein, a visualization analysis system was constructed for the real-time and specific localization of tumor regions by using the intratumorally overexpressed GSH in liver orthotopic transplantation tumors. Briefly, the slices from the tumor-bearing liver tissue were incubated with pFeSAN and chromogenic substrate (3,3′-Diaminobenzidine, DAB) for 6 min. After the incubation, the DAB substrate displayed brown oxidation product signal in the normal area of liver tissue section under the bright field microscope (Fig. 7f). However, due to the high GSH expression in tumor tissue, the oxidized DAB products were reduced by GSH into colorless DAB again, leading to no obvious brown signal in tumor area. Therefore, a clear distinction between tumor cells and adjacent health cells was observed in the para-carcinoma liver tumor tissue area, demonstrating that this pFeSAN-GSH assay methodology successfully distinguished tumor tissue from normal tissue. Next, the immunofluorescence staining as the gold standard was performed to verify the accuracy of this methodology (Fig. 7g). The expression of Ki67 protein as a marker in tumor pathology is strongly associated with tumor growth and proliferation[59,60]. The above results demonstrated that the intensity of the oxidized DAB decreased with the increase of tumor proliferation. In addition, Pearson's correlation coefficients of oxidized DAB (brown) and Ki67 (red) was -0.76, further indicating a high negative correlation (Fig. 7h). This was attributed to the oxidized DAB (brown)

being reduced to a colorless product with the increase of GSH in tumor tissues. Compared with fluorescence images of classic tumor proliferation marker (Ki-67) in adjacent sections of the same tumor-bearing liver tissue, the pFeSAN-GSH visualization detection exhibited a similar accuracy and a much shorter analysis time (6 min) than the conventional immunofluorescence diagnostic of tumor (>5 h) and avoided complex operation process as well[61]. All results demonstrated the facile, accessibility, and precision of the developed pFeSAN-GSH visualization analysis with great potentials for clinical tumor tissue detection.

Inspired by the structure of natural C$c$O, we demonstrated a biomimetic synthetic strategy for scalable synthesis of porous Fe-N₃ single-atom nanozymes using Hb as both template and Fe-source, which delivered a high oxidase-like activity. The mesoporous features of pFeSAN significantly promoted mass transport and maximumly exposed active iron sites during reaction. The Fe-N₃ moiety enabled a strong oxygen adsorption/activation, which underwent the complete O₂-to-H₂O pathway for oxidation, analogy with that of the natural C$c$O. Taking all advantages mentioned above, pFeSAN exhibited exceptional oxidase-like activity and durability. Based on the high oxidase-like performance for TMB oxidation and the reduction capability of GSH for oxTMB, the pFeSAN-GSH system was constructed for the colorimetric detection of GSH with high sensitivity, wide detecting range, and good specificity. Colorimetric analysis based on the pFeSAN assay was further developed for real-time and rapid detection of GSH in biological tissues and also demonstrated as the visualization methodology for the rapid and precise identifications of the tumor boundary. This work may open promising avenues for the design of single-atom nanozymes as alternatives of natural enzymes for biomedical applications.

## Methods

### Ethical statement

All animal experiments were approved by the Animal Experiment Administration Committee of the Fourth Military Medical University to ensure ethical and humane treatment of animals (KY2022593-1). All research was performed in accordance with relevant guidelines and regulations. The maximal tumor size/burden permitted by the Animal Experiment Administration Committee of the Fourth Military Medical University is 15 mm in diameter; this maximum size was not exceeded.

### Reagents

Hemoglobin (Hb) and glucose oxidase (GOx) were purchased from Shanghai Yuanye Bio-Technology Co. Ltd. $Fe_3O_4$, bovine serum albumin, sodium acetate (NaAc, >99%), glacial acetic acid (HAC, 99.99%), anhydrous dimethylsulfoxide (DMSO), iodosylbenzene (97%), and acetonitrile (99.9%) were purchased from Aladdin Co., Ltd. Heme, 3,3′,5,5′-tetramethylbenzidine, 2-methylimidazole, $Zn(NO_3)_2 \cdot 6H_2O$, superoxide dismutase, L-ascorbic acid, L-tryptophan, L-glycine, glutamic acid, mannitol, furfuryl alcohol, L-cysteine, L-arginine, glutathione, 5,5-dimethyl-1-pyrroline N-oxide, nafion solution (99%), methanol (99.9%), and 4′,6-diamidino-2-phenylindole (DAPI) were purchased from Sigma-Aldrich. Glutathione assay kit and 3,3′-Diaminobenzidine was purchased from Shanghai Beyotime Biotechnology Co., Ltd. Dulbecco's Modified Eagle Medium (DMEM), Dulbecco's Modified Eagle Medium/Nutrient Mixture F-12 (DMEM/F12), penicillin/streptomycin and fetal bovine serum (FBS) was obtained from Gibco. All aqueous solutions were prepared with deionized water (18.2 MΩ·cm, Millipore).

### Characterizations

Scanning electron microscope images were obtained from a Verios G4 ultra high-resolution field emission scanning electron microscope (FEI, USA). Transmission electron microscope images, elemental mapping, electron energy loss spectroscopy, and high-angle annular dark-field scanning transmission electron microscope (HADDF-STEM) images were performed on a Titan Themis G2 transmission electron microscope (FEI, USA) operated at 300 kV. Nanoparticle size distribution and zeta potential were measured by a Zetasizer Nano ZS90 nanoparticle size analyzer (Malvern, U.K.). Fourier transform infrared spectroscopy were obtained from a Nicolet iS20 Fourier transform infrared spectrometer (Thermo Scientific, USA). Thermogravimetric analysis was performed on a TGA8000 (PerkinElmer, USA) instrument from 30 °C to 1000 °C at a ramping rate of 5 °C $min^{-1}$ in the Ar. X-ray diffraction patterns were recorded by the D8 Advance X-ray diffractometer (Bruker, USA) by using CuKa radiation ($\lambda = 0.15418$ nm), the measurement had a step size of 0.02° over a range from 5 to 80°. X-ray photoelectron spectroscopy was conducted on a Thermo Scientific™ K-Alpha™+ spectrometer equipped with a monochromatic Al Kα X-ray source (1486.6 eV) operating at 100 W. Samples were analyzed under vacuum ($P < 10^{-8}$ mbar) with a pass energy of 150 eV (survey scans) or 50 eV (high-resolution scans). All peaks were calibrated with C1s peak binding energy at 284.6 eV for adventitious carbon. The Fe loadings were determined by inductively coupled plasma atomic emission spectrometry (Agilent 5110, USA). Raman spectra were explored with a Scientific LabRAM HR Evolution Raman spectrometer (HORIBA, Japan) equipped with a 532 nm laser. BET surface area, nitrogen adsorption and desorption isotherm linear plot and pore diameter distributions of various catalysts were observed by an automatic surface area and porosity analyzer (Micromeritics ASAP 2460, USA). The catalysts were degassed at 150 °C for 8 h and then tested for nitrogen adsorption and desorption. Ultraviolet and visible spectrophotometry absorption spectra were recorded using a TU-1950 UV-vis spectrophotometer in the wavelength range of 200–800 nm. The nitrogen vacancies were detected by an EMXplus 10/12 electron paramagnetic resonance spectrometer (Bruker, German). The electron paramagnetic resonance of pFeSAN

was measureed in acetonitrile/PhIO solution at 77 K to analyze its intermediate state, in which the experimental conditions were controlled at the frequency of 9.853 GHz, the microwave power of 20.0 mW, the center field of 3510 G, the modulation amplitude of 2.0 G and the time constant of 40.96 ms.

The extended X-ray absorption fine structure measurements of the catalysts were carried out the 21A X-ray nanodiffraction beamline of Taiwan Photon Source (TPS), National Synchrotron Radiation Research Center (NSRRC). This beamline adopted the 4-bounce channel-cut Si (111) monochromator for mono-beam X-ray nanodiffraction and X-ray absorption spectroscopy. The end-station equipped with three ionization chambers and Lytle/SDD detector after the focusing position of KB mirror for transmission and fluorescence mode X-ray absorption spectroscopy. The photon flux on the sample was range from $1 \times 10^{11}$ to $3 \times 10^{9}$ photon/sec for X-ray energy from 6 to 27 keV.

### Preparation of catalysts

The catalyst precursors were prepared by mixing 9 mL of methanol solution containing 2-methylimidazole (0.308 g) with 1 mL of aqueous Hb (60 mg/mL) or 1 mL of aqueous heme (2.2 mg/mL) or 1 mL aqueous BSA (1 mg/mL) or 1 mL of $H_2O$. Afterwards, 10 mL of methanol solution containing 0.279 g of $Zn(NO_3)_2 \cdot 6H_2O$ was added. After the continuous stirring for 6 h at room temperature, the solids were centrifuged (6000×$g$ for 5 min), and washed with methanol three times. The collected solids were dried at 60 °C to yield Hb@ZIF-8, Heme@ZIF-8, BSA@ZIF-8 and ZIF-8 as the precursors of respective catalysts. For synthesis of porous Fe single atom nanozymes (pFeSAN), the Hb@ZIF-8 precursors were placed in quartz boats in a tube furnace under a flowing of Ar (50 $cm^3$/min), raised up to the desired temperatures with a ramping rate of 5 °C $min^{-1}$, and then kept at the target temperatures for 3 h. The Fe single atom nanozymes (FeSAN), carbon supports (C) and porous carbon supports (pC) were also synthesized through the identical approach except with Heme@ZIF-8, BSA@ZIF-8, and ZIF-8 as precursors, respectively. For a comparison, Pt and $Mn_3O_4$ were prepared according to the previous literature[41,42].

### Oxidase-like activity

The oxidase-mimicking activity was evaluated by using TMB as substrate at 25 °C under acidic conditions (pH 4.0, 0.1 M HAc-NaAc buffer). The absorbance of oxTMB was recorded at 652 nm through TU−1950 UV-vis spectrophotometer. In detail, 20 μL of various catalysts aqueous solution (1 mg/mL) and 20 μL of TMB solution (50 mM in DMSO, DMSO is miscible with HAc-NaAc buffer) were sequentially added into HAc-NaAc buffer (0.1 M, pH 4.0). The final volume was fixed to 1 mL by adding deionized water (18.2 MΩ·cm). After 10 min, the catalytic oxidation of TMB was determined by their UV-vis absorption spectra.

To evaluate the activity of pFeSAN to reaction temperature and pH, its oxidase-like activity was measured at various temperatures (10, 20, 30, 40, 50, 60, and 70 °C) or medias with various pH values (2, 4, 6, 8, 10, and 12).

To evaluate the stability of pFeSAN, the catalysts were treated with 1 M HCl or 1 M NaOH solutions for 24 h, and then centrifuged off and washed with deionized water for 6 times before testing their oxidase-mimicking activity.

### Enzymatic kinetic analysis

Further kinetic experiments were performed for TMB oxidation at different concentrations by using various catalysts. The experiment conditions were similar to that of oxidase-like activity, in which 1.0 mL HAc-NaAc buffer (0.1 M pH 4) containing 20 μL of catalysts (1.0 mg/mL) aqueous solution and different volumes of TMB solution (50 mM DMSO). The kinetic constants ($K_M$ and $V_{max}$) were obtained according to Michaelis−Menten equation: $V = V_{max}[S]/(K_M + [S])$ (1), where V was the initial velocity, [S] was the concentration of the substrate, $K_M$ was

the Michaelis−Menten constant, and $V_{max}$ was the maximal reaction velocity. The value of $K_M$ was equivalent to the substrate concentration at which the rate of conversion was half of $V_{max}$. $V_{max}$ and $K_M$ were calculated from the Lineweaver-Burk double-reciprocal plot (1/V and 1/[S]). The absorbance signal was converted to concentration by the Beer-Lambert law (A = εbc) (2) where $\varepsilon = 39,000\,M^{-1}cm^{-1}$ at 652 nm for the oxidized TMB (oxTMB).

## Adsorption experiment

The relative difference in the TMB adsorption capability of pFeSAN and FeSAN was determined by the obvious UV-vis absorption peak of TMB at 285 nm. Firstly, the absorbance of different concentrations of TMB solutions in HAc-NaAc buffer (0.1 M, pH 7.0) were measured to plot a standard curve. Then 1.0 mL of NaAc solution (0.1 M HAc-NaAc buffer, pH 7.0) containing 50 μM TMB and 5 μg/mL catalysts were stirred for 10 min. Then, the supernatants were centrifuged off to measure the absorbance.

## Specific activity calculation

Calculate the nanozyme activity (units) using the following equation[42]:

$$b_{nanozyme} = V/(\varepsilon \times l) \times (\Delta A/\Delta t) \tag{3}$$

where $b_{nanozyme}$ is the catalytic activity of nanozyme expressed in units. One unit is defined as the amount of nanozyme that catalytically produces 1 μmol of product per min at room temperature; V is the total volume of reaction solution (μL); $\varepsilon$ is the molar absorption coefficient of the colorimetric substrate, which is maximized at $39,000\,M^{-1}\,cm^{-1}$ at 652 nm for TMB; l is the path length of light traveling in the cuvette (cm); A is the absorbance after subtraction of the blank value; and $\Delta A/\Delta t$ is the initial rate of change in absorbance at 652 nm $min^{-1}$.

Calculate the specific activity of the nanozyme (U $mg^{-1}$) by

$$a_{nanozyme} = b_{nanozyme}/[m] \tag{4}$$

where $a_{nanozyme}$ is the specific activity expressed in units per milligram (U $mg^{-1}$) nanozymes, and [m] is the nanozyme weight (mg) of each assay.

## Catalytic mechanism of pFeSAN

To verify the dependence of the oxidase-like activity of pFeSAN on oxygen, taking it as a control in air, nitrogen, or oxygen was introduced into the reaction flask for 30 min before measuring oxidase-like activity. Subsequently, the absorbance was measured after 10 min of sealing reaction.

The influences of various scavengers on the oxidase-like activity of pFeSAN were investigated by comparing the activity in the absence or presence of ROS scavengers. In the catalytic oxidation of TMB, the pFeSAN was added firstly into the HAc-NaAc buffer (pH 4.0) containing SOD (0.5 mg/mL) or mannitol (50 mM) or FFA (5 mM) and TMB (1 mM). Then, the absorbance at 652 nm was measured respectively.

The free radicals (such as •OH and •$O_2^-$) in reaction systems were examined by a Bruker spectrometer in the HAc-NaAc (pH 4 or pH 7) buffer, using DMPO as the spin-trapping agent.

## Electrochemical analysis

The catalyst ink was prepared by dispersing the nanozymes (4 mg) through sonication in a mixed solvent (1 mL) containing 32 μL of a 5.0 wt% Nafion solution, 200 μL of ethanol and 768 μL of deionized water. The cyclic voltammetry (CV) curves were evaluated using a three-electrode system on an electrochemical workstation (CHI 760E, Shanghai Chenhua, China), including the carbon paper (CP) as the working electrode, a graphite rod as the counter electrode, and a saturated calomel electrode (SCE) as the reference electrode. Briefly, 10 μL of the ink was loaded onto CP with a catalyst loading of 0.56 mg $cm^{-2}$. The CV tests were performed in an $O_2$-saturated NaAc solution (0.1 M, pH 4.0) with TMB (1 mM). All potential values were calibrated to the reversible hydrogen potential ($E_{RHE}$) ($E_{RHE} = E_{SCE} + 0.241 + 0.0592 \times pH$). The CV measurements were recorded at a scan rate of 50 mV $s^{-1}$ in a potential range of −0.2–1.2 V.

The impedance spectra were measured on a PGSTAT302N AUTOLAB (Metrohm, Switzerland) at the voltage frequency ranging from $10^5$ Hz to $10^{-1}$ Hz and the applied potential of −0.1 V.

To verify the electron transfer number, an aliquot (20 μL) of the ink was then dropped on a rotating ring disk electrode (RRDE) glassy carbon electrode with a diameter of 4 mm and a loading of 0.64 mg $cm^{-2}$. The RRDE measurements were carried out at a rotating speed of 1200 rpm. The hydrogen peroxide ($H_2O_2$) yield (%) and the electron transfer number (n) can be calculated based on the Equation:

$$H_2O_2(\%) = 200 I_D/N(I_D + I_R/N) \tag{5}$$

$$n = 4 I_D/(I_D + I_R/N) \tag{6}$$

where $I_D$ is the disk current, $I_R$ is the ring current, and N is the collection efficiency of the ring electrode (0.39 in this work)[17].

## Density functional theory calculation

All the spin-polarized DFT calculations are performed by the Vienna Ab initio Simulation Package (VASP5.4.1 + VTST) with the projector augmented wave (PAW) method[62]. The exchange-functional is treated using the generalized gradient approximation (GGA) with Perdew-Burke-Ernzerhof (PBE) functional. The energy cutoff for the plane wave basis expansion was set to 400 eV. Partial occupancies of the Kohn−Sham orbitals were allowed using the Gaussian smearing method and a width of 0.2 eV. The single-layer graphene with the active center of Fe-$N_4$ was built, where one of the coordinated N was removed to build the structure of Fe-$N_3$. The k-point of $2 \times 2 \times 1$ was used in the Brillouin zone for all surface structure optimization. The self-consistent calculations apply a convergence energy threshold of $10^{-5}$ eV, and the force convergency was set to 0.05 eV/Å. The reaction free energy was calculated following the computational hydrogen electrode (CHE) model. The free energy corrections were considered at the temperature of 298 K, following:

$$\Delta G = \Delta E + \Delta GZPE + \Delta GU - T\Delta S \tag{7}$$

where ΔE, ΔGZPE, ΔGU, and ΔS refer to the DFT calculated energy change, the correction from zero-point energy, the correction from inner energy, and the correction from entropy[63].

The solvent effect was considered due to the stabilization of adsorbate from the H-bond network in the. A stabilization of −0.17, and −0.20 eV were considered for OH*, and OOH* according to previous study[64].

## Colorimetric detection of GSH based on the pFeSAN

In the standard procedure, the assays were performed in 1 mL HAc-NaAc buffer (0.1 M, pH 4.0) with different concentrations of GSH (0.05, 0.5, 2.5, 5, 12.5, 25, 50, 100, 150, 200, 250, 500, or 1000 μM), where the TMB and pFeSAN concentrations were 1 mM and 20 μg/mL, respectively. After 10 min of incubation at room temperature, a standard curve was established by measuring the UV-vis absorbance of solutions.

## Anti-interference evaluation

To explore anti-interference of the GSH detection system, various ions and molecules were independently mixed and then the peak intensity at 652 nm was measured after 10 min of incubation. Specifically, 20 μL of pFeSAN (1 mg/mL) and 20 μL of TMB (50 mM) were added into

HAc-NaAc buffer (0.1 M, pH 4.0), which contains 0.1 mM GSH, Na⁺, Ca²⁺, Mg²⁺, K⁺, Glu, Trp, Arg, Gly, Cys, L-AA, glucose, GOx or BSA. The final volume was fixed to 1 mL by adding deionized water. After reacting at room temperature for 10 min, the activity of pFeSAN were recorded in the presence of various ions or molecules according to the standard procedure as mention above.

### Effect of GSH on the pFeSAN
The pFeSAN was incubated with GSH (1 mM) for 10 min and then collected by centrifugation and thorough washing by pure water for three times to obtain the GSH-pretreated pFeSAN. To analyze the effect of GSH on the activity of the pFeSAN nanozyme, 20 μL of pFeSAN or GSH-pretreated pFeSAN (1 mg/mL), 20 μL of TMB (50 mM) and 960 μL of HAc-NaAc buffer (pH 4.0) were mixed and incubated at 25 °C for 10 min. The absorbance of each reaction solution at the wavelength of 500–800 nm was recorded.

### Detection of GSH in cell samples
Hepa 1–6 cells (CL-0105, Procell) were cultured in DMEM medium (10% FBS and 1% penicillin/streptomycin) at 37 °C in a humidified atmosphere containing 5% CO₂. AML-12 cells (CL-0602, Procell) were grown in DMEM/F12 medium with 10% FBS and 1% penicillin/streptomycin at 37 °C. The cell lines were genetically authenticated by the supplier based on Short Tandem Repeat method. To prepare Hepa 1–6 and AML-12 cell samples, the cells were digested by trypsin-ethylenediaminetetraacetic acid and then re-suspended in phosphate-buffered saline (10 mM PBS, pH = 7.4).

Then, 20 μL of pFeSAN (1 mg/mL) and 20 μL of TMB (50 mM DMSO) were sequentially added into HAc-NaAc buffer (0.1 M pH 4) and the final volume was fixed to 1 mL, which contained different amounts of Hepa 1–6 or AML-12 (1000, 10,000, 50,000, 100,000, 200,000, 300,000, 500,000 or 800,000). After 10 min, the catalytic oxidation of TMB was investigated by the UV-vis absorption spectra.

### Mice and tumor-bearing mouse model
C57BL/6 mice (male, 7 weeks) were maintained in a specific pathogen-free facility. All animal experiments were approved by the Animal Experiment Administration Committee of the Fourth Military Medical University to ensure ethical and humane treatment of animals (KY2022593-1). For orthotopic liver tumor models, the Hepa1–6 cells (5 × 10⁶) were suspended in 25 μL of Matrigel (Sigma-Aldrich) and inoculated into the liver parenchyma of the left lobe for in situ liver tumor model (n = 6). Whereafter, C57BL/6 mice were euthanized for three weeks after inoculation, and part of the tumor-bearing liver tissue for immunofluorescence staining and intrahepatic GSH detection.

### Intrahepatic GSH detection
In order to analyze the GSH distribution within the tumor-bearing liver tissue, the liver samples were subjected to the frozen section and then stained by pFeSAN and DAB for 5 min. Then, the stained tissues were monitored by using an inverted microscope.

### Immunofluorescence staining
The tumor-bearing liver tissues from C57BL/6 mice were frozen section, and then the slices were permeabilized with 0.3% Triton X-100 solution for 15 min at room temperature. After blocking the slices with 5% BSA for 30 min at 37 °C, tumor-bearing liver tissue slices were incubated with Ki67 antibody (1:250, Abcam, ab16667) overnight at 4 °C. Afterwards, the slices were washed for three times with PBS and stained with secondary antibody Goat Anti-Rabbit IgG-Alexa Fluor 594 (1:200, Abcam, ab150080) for 1 h at 25 °C. Finally, the tissue sections were washed thoroughly with PBS, counterstained with DAPI and imaged under a laser scanning confocal microscope (FV-1000, Olympus, Japan).

### Statistical analysis
The data were analyzed using Origin software (version 2018). All experiments were repeated at least 3 times and presented as mean ± SD. Asterisks indicate significant differences (**$p < 0.01$), analyzed by the Student's two-tailed test.

### Reporting summary
Further information on research design is available in the Nature Portfolio Reporting Summary linked to this article.

## Data availability
All data generated that support the findings of this study are present in the main text and the Supplementary Information file. Source data are provided with this paper.

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

## Acknowledgements

We acknowledge the financial support from the National Natural Science Foundation of China (Z.T., 52002314 and Y.Q., 21872109). The authors also acknowledge the support from Fundamental Research Funds for the Central Universities (Z.T., D5000210635 and Y.Q., D5000210829). The calculations were supported by TianHe-2 at Shaanxi Supercomputing Center of China and Central for High Performance Computing of Northwestern Polytechnical University.

## Author contributions

Z.T. and Y.Q. conceived and designed the project. D.C., Z.X., Z.G., W.G., X.T., and W.L. performed the experiments. Z.T. and Y.Q. supervised the project. Z.T., J.Z., X.Z., and S.Z. analyzed data. D.C., Z.T., and Y.Q. wrote and edited the manuscript. All authors discussed the results and contributed to the preparation.

## Competing interests

The authors declare no competing interests.
