## [Peer Review File · Nature Communications]

Reviewers' Comments:

Reviewer #1:

Remarks to the Author:

This paper presents an innovative biomimetic synthetic strategy for the scalable synthesis of porous Fe-N₃ single atom nanozymes (pFeSAN) using hemoglobin as a template. The authors demonstrate the superior catalytic activity of pFeSAN compared to Fe-N₄ and Fe₃O₄ nanozymes, attributing it to the suppressed aggregation of atomically dispersed Fe, facilitated mass transfer, and maximized exposure of active sites. Additionally, pFeSAN was successfully applied in the rapid colorimetric detection of glutathione and developed as a real-time, facile, rapid, and precise visualization analysis methodology for tumors via glutathione levels. Overall, this paper is a valuable contribution to the field of nanomaterials with potential applications in diagnostics and therapeutics. Therefore, I recommend it can be accepted by Nature Communications after revision if some problems can be properly addressed:

1. Figure 5f: the EPR patterns of the high-valent iron-oxygen species formed by SACs with different coordination numbers (Fe-N_x) should be different. In this manuscript, the number of N atoms coordinated with Fe is 3. Why is the EPR signal of the high-valent Fe oxygen species formed by FeN₃ the same as the signal formed by FeN₅ (Refs. 21 and 23)?
2. Page 10, in the XANES results, it should be "The rising-edge position of the X-ray absorption near-edge structure spectra (XANES) of pFeSAN was between those of Fe₂O₃ and Fe foil (not FePC), and very close to that of FePc."
3. Page 11, SACs can be regarded as a complex (coordination compound). The chemical configuration (such as the coordination number, the chemical environment of the coordination atom, etc.) of the active site in SACs will change the electronic structure of the metal by affecting the splitting energy and stability energy of the central d orbital of the metal. Why the author judged that the electronic structure around Fe in FeN₃ is similar to that in FePC through the partial overlap of absorption edges? In addition, the author did not comprehensively consider the Edge front peak and white line peak.
4. Compared with R-space, the data in k-space can better reflect the quality of synchrotron radiation data. So, the real data and fitting data in k-space should be included.
5. For R-space, the data after 4 Å is meaningless. In order to better show the R-space data and the situation, it is recommended to shorten the Abscissa to less than 6 Å (Figures 3b and 3c). In addition, the Abscissa of R-space should be R (Å) only after the phase shift is corrected. I am wondering whether no phase shift correction was performed for the data in the figure, if it is, the Abscissa should be R + α (Å), please check it.
6. It is mentioned by the author that Hb is competitive in price. But the price of Hb (US \$3.60 / g, Shanghai Yuanye, S12021-5g) is significantly higher than that of FeCl₃.
7. Lack of comparison, compared with other nano-enzymes with oxidase-like activity, what is the advantage of pFeSAN in the kinetic constant?
8. For Figure 5a and 5b, the author pointed out that "no apparently detectable signals of any ROS further proved that ROSs were not the main intermediates for the oxidase-like activity of pFeSAN, which was very different from the majority of the previously reported nanozymes", please give more discussion on this phenomenon.
9. For Figure 2f, the resolution is too low.

Reviewer #2:

Remarks to the Author:

Due to the high atom utilization and remarkable catalytic activity, single-atom catalysts exhibit great potentials for applications of nanozyme. In this work, a porous Fe single-atom catalyst prepared by a biomineralization–pyrolysis strategy exhibited remarkable oxidase-like activity, which was used for GSH sensing and cancer cell identification. The porous Fe single-atom nanozyme was well characterized. And, the mechanism of oxidase-like activity was investigated as well, which opened up a new way to design and optimize nanozymes. Therefore, this manuscript can be accepted to be published by Nature Communications after proper revisions.

1. Fe single-atom nanozyme reported in the past often exert catalase-like catalytic activity. In this study, does pFeSAN have catalase-like activity?

2. Compared with previously reported Fe single-atom nanozyme, what are the advantages of pFeSAN in structure-activity characteristics?
3. The label of elements and scale bar at Supplementary Fig. 11 and 15 was not clearly visible.
4. Please provide the quantitative analysis of EIS (Fig. 5d).
5. In this work, did the pFeSAN show obvious toxicity to normal cells and tumor cells, resulting in the influenced cellular GSH analysis?

Reviewer #3:

Remarks to the Author:

This paper deals with the development of a porous 3-coordinate iron-based nanozyme which can exhibit oxidase-like activity. The authors use hemoglobin as Fe source/template and incorporate it into a zeolitic imidazolate framework (ZIF-8). The single-atom nanozyme called pFeSAN acts as an oxidase model by oxidizing the colorimetric substrate TMB to oxTMB, which helps in the detection of glutathione (GSH) in nM concentrations. The authors show a significant improvement in the oxidase-like activity; the nanozyme exhibiting 3.3- and 8791-fold higher oxidase-like activity than the Fe-N4 and Fe3O4 nanozymes reported earlier. As the GSH level is altered in many diseases, including cancer, the authors propose that the nanozyme with oxidase-like activity can be used for the visualization of tumor. The work is interesting, but does not warrant publication in *Nat. Commun.* due to the following reasons.

(1) Development of Fe-based single-atom nanozyme by incorporating hemoglobin into ZIF-8: The topic of Fe-based single atom nanozyme is not new. It has been shown that bovine hemoglobin can be incorporated into ZIF-8 to develop single atom nanozymes (*ACS Appl. Mater. Interfaces* 2016, 8, 29052–29061). This nanozyme has been used as peroxidase mimic to oxidize TMB for the colorimetric detection. Many other Fe-based single atom nanozymes with oxidase-like activity have been reported in a review (*Nano Res.* 2023, 16, 1992–2002). The mechanism of the structure-dependent oxidase-like activity of Fe-N/C nanozyme has also been reported (*Chem. Commun.* 2019, 55, 5271–5274).

(2) Single atom nanozymes for the detection of glutathione (GSH). Nanozymes and Fe-N-C-based single atom nanozymes have been used extensively for the detection of glutathione (*Journal of Materiomics*, 2022, 6, 1251-1259). Mn3O4 microspheres have been used as an oxidase mimic for rapid detection of glutathione (*RSC Adv.*, 2019, 9, 16509-16514). Light-responsive MOF as an oxidase mimic for cellular GSH detection (*Anal. Chem.* 2019, 91, 13, 8170–8175). MnO2 nanosheets as an artificial enzyme to mimic oxidase for rapid and sensitive detection of glutathione (*Biosensors & Bioelectronics*, 2017, 90, 69-74). Therefore, the present study lacks novelty in the detection of glutathione.

(3) The concept of using GSH as biomarker for visualization of cancer cells is not entirely new. There are many recent reports in the literature which highlight the concept. For example, a MOF has been reported to exhibit oxidase-like activity by oxidizing TMB to oxTMB, which has been used as a colorimetric probe for GSH detection. The oxidase mimic has been used to analyze the GSH level in the lysates of normal and cancer cells (*Anal. Chem.* 2019, 91, 8170–8175). There are many other reports which describe the use of nanozymes for tumor visualization through GSH detection.

Point-by-point Response to Reviewer(s)' Comments

Reviewer #1 (Remarks to the Author):

This paper presents an innovative biomimetic synthetic strategy for the scalable synthesis of porous Fe-N₃ single atom nanozymes (pFeSAN) using hemoglobin as a template. The authors demonstrate the superior catalytic activity of pFeSAN compared to Fe-N₄ and Fe₃O₄ nanozymes, attributing it to the suppressed aggregation of atomically dispersed Fe, facilitated mass transfer, and maximized exposure of active sites. Additionally, pFeSAN was successfully applied in the rapid colorimetric detection of glutathione and developed as a real-time, facile, rapid, and precise visualization analysis methodology for tumors *via* glutathione levels. Overall, this paper is a valuable contribution to the field of nanomaterials with potential applications in diagnostics and therapeutics. Therefore, I recommend it can be accepted by Nature Communications after revision if some problems can be properly addressed:

We thank the reviewer for his/her constructive comments on our manuscript. We would like address his/her comments as below.

1. Figure 5f: the EPR patterns of the high-valent iron-oxygen species formed by SACs with different coordination numbers (Fe-N_x) should be different. In this manuscript, the number of N atoms coordinated with Fe is 3. Why is the EPR signal of the high-valent Fe oxygen species formed by FeN₃ the same as the signal formed by FeN₅ (Refs. 21 and 23)?

Response:

Thank you for this constructive comment. Fe-heme structure of natural cytochrome c oxidase (CcO) plays the core roles for the O₂ activation through the O₂-to-H₂O pathway, delivering high catalytic activity for many oxidation reactions. The catalytic cycle of CcO for oxygen reduction starts with the binding of O₂ on a ferrous heme to afford a Fe(IV)=O intermediate, followed by the reduction of O₂ into H₂O (Ref. 45: *Acc. Chem. Res.*, 2007, 40, 7, 554). The Fe(IV)=O is considered as the key intermediate in the catalytic cycles of many heme iron enzymes and heme analogs for oxidation. Although

the coordination environments of Fe-N₃ and Fe-N₅ are not identical, they share the common mechanistic grounds through the Fe(IV)=O intermediate (Ref. 45: *Acc. Chem. Res.*, 2007, 40, 7, 554). In this work, a typical diamond-shaped label signal at g = 2.03 was detected by EPR, consistent with the η²-peroxo heme species, signifying the formation of Fe(IV)=O intermediate. This suggests that Fe-N₃ active center of pFeSAN may have a similar reaction process with heme analogs. We suggest that the same oxidase-like reactivities and Fe(IV)=O intermediate of Fe-N₃ and Fe-N₅ structure are mainly caused by their active site structures, but not the axial ligand. The more information has been added in the revised manuscript (Page 20, Line 3-6).

2. Page 10, in the XANES results, it should be “*The rising-edge position of the X-ray absorption near-edge structure spectra (XANES) of pFeSAN was between those of Fe₂O₃ and Fe foil (not FePC), and very close to that of FePc.*”.

Response:

Thank you sincerely for your suggestion. In the revised manuscript, we have modified the related information “*The rising-edge position of the X-ray absorption near-edge structure spectra (XANES) of pFeSAN was between those of Fe₂O₃ and Fe foil, and very close to that of FePc,*” in the main context (Page 10, Line 19-21).

3. Page 11, SACs can be regarded as a complex (coordination compound). The chemical configuration (such as the coordination number, the chemical environment of the coordination atom, etc.) of the active site in SACs will change the electronic structure of the metal by affecting the splitting energy and stability energy of the central d orbital of the metal. Why the author judged that the electronic structure around Fe in FeN₃ is similar to that in FePC through the partial overlap of absorption edges? In addition, the author did not comprehensively consider the Edge front peak and white line peak.

Response:

Thank you very much for pointing this out. XANES reveals slight differences between the XANES spectra of pFeSAN and FePc. The latter shows a pre-edge peak at 7118 eV

assigned to a $1s \rightarrow 4p_z$ shakedown transition characteristic for a square-planar configuration with high D_{4h} symmetry (Figure R1). For characteristics of pFeSAN, the pre-edge feature is absent for pFeSAN, revealing a broken D_{4h} symmetry (Ref. 38: *Nat. Mater.*, 2015, 14, 937–942). The theoretical XANES was also simulated with FDMNES code (*J. Phys.: Condens. Matter*, 2009, 21, 345501). The optimized Fe–N₃ structure was used as the input structure. The sphere radius to calculate the cluster adsorption was 12 Å. The result showed that the relative pre-edge and white line peaks of the simulated spectrum were consistent with the experiment, validating the Fe–N₃ local structure of pFeSAN (Figure R2).

Figure R1. XANES of Fe foil, FePc, Fe₂O₃ and pFeSAN.

Figure R2. Comparison of the experimental XANES curve (pFeSAN) with the calculated XANES data of Fe–N₃. The insets show the optimized Fe–N₃ structure. The relative pre-edge and white line peak of the simulated spectrum are consistent with the experiment.

The relative information has been updated in the revised manuscript (Page 11, Line 1–4 and 16–19, and Supplementary Fig. 19).

4. Compared with R-space, the data in k-space can better reflect the quality of synchrotron radiation data. So, the real data and fitting data in k-space should be included.

Response:

Thank you for your careful review. The best-fit analysis is shown in Figure R3. The dominant contribution is given by Fe–N first shell coordination. In the revised manuscript, we have added the related information in the main context (Page 11, Line 9–11 and Fig. 3d).

Figure R3. EXAFS fitting results for Fe foil, FePc, Fe_2O_3 and pFeSAN at k-space.

5. For R-space, the data after 4 \AA is meaningless. In order to better show the R-space data and the situation, it is recommended to shorten the Abscissa to less than 6 \AA (Figures 3b and 3c). In addition, the Abscissa of R-space should be R (\AA) only after the phase shift is corrected. I am wondering whether no phase shift correction was performed for the data in the figure, if it is, the Abscissa should be $R + \alpha$ (\AA), please check it.

Response:

We appreciate the reviewer's suggestion. We changed the abscissa according to the reviewer's suggestion and updated Figures 3b and 3c in the revised manuscript (Page 12, Fig. 3b and 3c). Meanwhile, the wavelet transform used the raw data of XAFS directly, no phase shift correction was performed.

6. It is mentioned by the author that Hb is competitive in price. But the price of Hb (US \$3.60/g, Shanghai Yuanye, S12021-5g) is significantly higher than that of FeCl₃.

Response:

We appreciate the reviewer's valuable comments. The prices of Hb and FeCl₃ are greatly depended on their grades and suppliers. Herein, we provided the prices of two iron resources for reference. At least, the price of Hb is accepted as the cheap resource. The price of Hb is US \$42.7 per 100 g from Energy Chemical, D110156-100g and the FeCl₃ is US \$50.3 per 100 g from Sigma-Aldrich (157740-100G), whereas the activity of Hb-templated pFeSAN can be up to 8791 times higher than that Fe₂O₃ with FeCl₃ as iron source. Therefore, this one-pot biomimetic synthetic strategy is efficient and low-cost. We agree the Reviewer's comments and made the change in the revised manuscript in Page 13, Line 6–7.

Besides, blood protein (hemoglobin and albumin) as a biomass, which is normally discarded as waste and causes serious environmental pollution (*Adv. Mater.*, 2018, 30, e1703691). This work offers a promising green route to attain high-value catalysts from biomass and demonstrates another advantage of using Hb as an iron source and template for iron single atom catalysts.

More importantly, Hb has three significant advantages over FeCl₃:

- (1) Fe ions residing in the structure of Hb could be isolated by protein structure, effectively preventing the aggregation of atomically dispersed iron active sites during harsh pyrolysis process;
- (2) Hb with a size of 2~3 nm forms a large number of mesopores during pyrolysis, benefiting the formation of mesopores and the maximum exposure of atomic Fe sites and also facilitating mass transfer during catalysis;
- (3) Unsymmetrically coordinated Fe–N₃ active site of pFeSAN was prepared by the Hb-pyrolysis strategy, delivering elevated oxidase-like activity.

Oppositely, FeCl₃ as a Fe source requires careful regulation to prevent metal site from agglomeration. Also, other hard templates are needed to create the mesoporous

structures, requiring the precise control of the operation. In contrast, our synthesis strategy is facile and easily scale-up.

7. Lack of comparison, compared with other nano-enzymes with oxidase-like activity, what is the advantage of pFeSAN in the kinetic constant?

Response:

We appreciate the reviewer’s valuable suggestions. Besides the control catalysts of Fe₃O₄ and FeSAN (Fe–N₄), we also systematically compare the performance of pFeSAN with the previously reported Pt NPs (Ref. 41: *Biosens. Bioelectron.* 2017, 92, 442) and Mn₃O₄ (Ref. 42: *Sens. Actuators B Chem.* 2021, 333, 129560) on their oxidase-like activity using TMB as a chromogenic substance. The relative activities of the pFeSAN were **5882** and **1176** times higher than that of Pt and Mn₃O₄, respectively, indicating that the oxidase-like activity of the pFeSAN was much higher than Pt and Mn₃O₄ (Figure R4a). According to Figure R4b,c, pFeSAN has a higher affinity for substrate TMB than Pt and Mn₃O₄, and the catalytic reaction rate is much higher than both Pt and Mn₃O₄. The relative information has been updated in revised manuscript (Page 16, Line 20–22 and Supplementary Fig. 31).

Figure R4. Oxidase-like activity evaluation. a. Comparison of relative oxidase-like catalytic activities of different nanozymes. b. Comparison of kinetics for pFeSAN, Pt and Mn₃O₄. K_m is the Michaelis-Menten constant. V_{max} is the maximal reaction velocity. c. Steady-state kinetic assay of Pt and Mn₃O₄ with TMB as substrate.

8. For Figure 5a and 5b, the author pointed out that “no apparently detectable signals of any ROS further proved that ROSs were not the main intermediates for the oxidase-like activity of pFeSAN, which was very different from the majority of the previously reported nanozymes”, please give more discussion on this phenomenon.

Response:

Thank you very much for your comments. Under N₂ atmosphere, TMB could not be oxidized by pFeSAN. Oppositely, the oxidase-like activity of pFeSAN could be boosted up to 2.2-fold by increasing the O₂ concentration to saturation (Figure 5a). Therefore, the high O₂-dependent activity of pFeSAN for substrate oxidation suggests the essence of O₂ as oxidants for catalytic reaction. Meanwhile, ROSs were undetectable by using either ROS quenchers (Supplementary Fig. 32a) and EPR spectra (Supplementary Fig. 32b). These results strongly confirmed that pFeSAN mediated the complete reduction of O₂ to H₂O without the release of free ROSs. This phenomenon verified that pFeSAN likely followed an oxygen atom transfer mechanism, similar to that of the natural *CcO* enzymes mediated by the Fe(IV)=O intermediate, which was very different from the majority of the previously reported oxidase-like nanozymes through the free ROS pathway (Ref. 23: *Anal. Chem.* 2022, 94, 15270; Ref. 43: *Catal. Sci. Technol.* 2021, 11, 7255). The Fe(IV)=O is considered a key intermediate in the oxidase-like reaction of many heme iron enzymes and heme analogs. (Ref. 45: *Acc. Chem. Res.*, 2007, 40, 7, 554). We speculated that the formation of Fe(IV)=O intermediates enabled the efficient oxidation of TMB substrate into ox-TMB. In the revised manuscript, we have added the related information in the main context (Page 18, Line 6–8).

9. For Figure 2f, the resolution is too low.

Response:

Thank you for your kind suggestions. In the revised manuscript, we have updated the related information in the Results section (Page 7, Fig. 2f).

Reviewer #2 (Remarks to the Author):

Due to the high atom utilization and remarkable catalytic activity, single-atom catalysts exhibit great potentials for applications of nanozyme. In this work, a porous Fe single-atom catalyst prepared by a biomineralization–pyrolysis strategy exhibited remarkable oxidase-like activity, which was used for GSH sensing and cancer cell identification. The porous Fe single-atom nanozyme was well characterized. And, the mechanism of oxidase-like activity was investigated as well, which opened up a new way to design and optimize nanozymes. Therefore, this manuscript can be accepted to be published by Nature Communications after proper revisions.

We thank the reviewer for his/her useful suggestions on our manuscript. The point-to-point response to the reviewers' comments is shown as below.

1. Fe single-atom nanozyme reported in the past often exert catalase-like catalytic activity. In this study, does pFeSAN have catalase-like activity?

Response:

Thank you very much for this constructive comment. Our experimental results show that the pFeSAN possesses ideal catalase-like (CAT-like) activity under neutral conditions (pH 6.0-7.0), similar to those previously reported studies (*Chem. Commun.*, 2019, 55, 14534-14537; *Adv. Mater.*, 2022, 34, e2205324; *Nat. Commun.*, 2022, 13, 4744.) (Figure R5a,b). Besides, the elimination efficiency of H₂O₂ was dependent on the concentration of pFeSAN (Figure R5c). The CAT-like activity of pFeSAN may provide opportunities for its in biomedical applications, such as the treatments of neurodegenerative diseases, inflammation, as well as cancer treatment.

Figure R5. a. The photo of comparison of the mixed solution H₂O₂ in the absence and presence of pFeSAN with the concentration of 40 μg/mL. b. Comparison of CAT-like activities of pFeSAN at various pH (4–12) conditions. c. The H₂O₂ scavenging of pFeSAN with different concentrations.

2. Compared with previously reported Fe single-atom nanozyme, what are the advantages of pFeSAN in structure-activity characteristics?

Response:

We thank the Reviewer for the positive comments. Developing artificial enzymes with the excellent catalytic performance of natural enzymes has been a long-standing goal for chemists. Fe single-atom catalysts coordinated with N atoms (Fe-N-C) exhibit well-defined atomic structures and electronic coordination environments and thereby deliver various enzyme-like catalytic activity. Unfortunately, majority of them were synthesized *via* pyrolysis at high temperatures, leading to structural collapse and part of the buried Fe-N_x units inaccessible to biomolecules. Also, the strong stacking of those N-doped graphite carbon in various Fe-N-C nanozymes generally induces the frustrated diffusion of bio-substrates to metal sites. Hence, various methods including spatial confinement, defect/vacancy engineering and coordination modulations have been developed to solve those problems. However, only part of those inadequacies could be overcome. Thus, seeking for a new synthetic strategy of single-atom metal nanozymes to simultaneously achieve the atomic metal dispersion, modulated electronic structure, elevated mass transport and tailorable coordination environment is still on high demands.

In this work, we report a straightforward Hb-mediated approach to efficiently distribute iron atoms homogeneously and in an atomically dispersed fashion across the nitrogen-doped carbon support. Meanwhile, the as-prepared three-coordinated single-atom Fe nanozyme possesses significantly higher specific surface area and mesoporosity to facilitate the mass transport and exposure of active Fe sites, and exhibits ideal oxidase-like catalytic activity. The benefits of our approach could be described as below.

- (1) pFeSAN with the unsymmetrically coordinated Fe-N₃ active sites was synthesized successfully by using Hb as Fe source. This synthesis represented a simple, facile and scalable one.
- (2) Evenly distributed Fe atoms in Hb effectively avoided the agglomeration of active sites during pyrolysis and created mesoporous structure (3~4 nm) in the pFeSAN, thereby maximumly exposing the atomic Fe sites and

significantly facilitating the mass transfer of reactants/products during the catalytic process.

- (3) pFeSAN delivered outstanding oxidase-like activity, which was 3.3- and 8791- times higher than those of Fe-N₄ and Fe₃O₄ nanozymes, respectively.
- (4) Mechanism investigations illustrated that pFeSAN underwent a catalytic pathway of the four-electron reduction of O₂ into H₂O, being identical to that of CcO, which was very different from the majority of the previously reported oxidase-like nanozymes with the generation of ROS.
- (5) pFeSAN as a highly-performed nanozyme exhibited a much higher upper detection limit of GSH at 1 mM, which was 2.5 to 40-fold higher than those of the previously reported investigations (Table R4).
- (6) Visual and rapid detection of tumor tissues through GSH colorimetric analysis was achieved for the first time, which was expected to help the effective resection of tumor tissues.

3. The label of elements and scale bar at Supplementary Fig. 11 and 15 was not clearly visible.

Response:

We are grateful for the suggestion. We have added the label of elements and scale bar in the revised Supplementary Information Fig. 11 and 15.

4. Please provide the quantitative analysis of EIS (Fig. 5d).

Response:

Thank you for your insightful comments. According to the EIS fitting results, it is known that the material resistance of pFeSAN is much smaller than that of FeSAN. Figure 5d shows the Nyquist plots of pFeSAN and FeSAN, where the diameter of the semicircle is associated with the electron-transfer resistance (R_{ct}). The R_{ct} decreased substantially from 407.3 to 298.6 Ω , implying enhanced electron transfer property and better mass transfer performance, which were consistent with the above results from CV (Fig. 5c). We have revised the manuscript and added the description in Results section (Page 19, Line 13).

5. In this work, did the pFeSAN show obvious toxicity to normal cells and tumor cells, resulting in the influenced cellular GSH analysis?

Response:

As suggested by the reviewer, we have now performed a biosafety analysis *in vitro*. In order to make sure the biosafety of pFeSAN, we examined the biocompatibility of the pFeSAN through in vitro cell viability. As shown in Figure R6, pFeSAN exhibited negligible cytotoxicity (cell viability >90%) on normal liver cells (AML12) and liver tumor cells (Hepa 1-6) even at a high concentration of 200 $\mu\text{g}/\text{mL}$, corroborating the high biocompatibility of the pFeSAN.

Figure R6. Cellular viability of AML12 and Hepa 1-6 cells treated with various concentrations of the pFeSAN for 3 h.

Reviewer #3 (Remarks to the Author):

This paper deals with the development of a porous 3-coordinate iron-based nanozyme which can exhibit oxidase-like activity. The authors use hemoglobin as Fe source/template and incorporates it into a zeolitic imidazolate framework (ZIF-8). The single-atom nanozyme called pFeSAN acts as an oxidase model by oxidizing the colorimetric substrate TMB to oxTMB, which helps in the detection of glutathione (GSH) in nM concentrations. The authors show a significant improvement in the oxidase-like activity; the nanozyme exhibiting 3.3- and 8791-fold higher oxidase-like activity than the Fe-N₄ and Fe₃O₄ nanozymes reported earlier. As the GSH level is altered in many diseases, including cancer, the authors propose that the nanozyme with oxidase-like activity can be used for the visualization of tumor. The work is interesting, but does not warrant publication in Nat. Commun. due to the following reasons.

We thank the Reviewer for raising the critical and constructive comments on our manuscript. According to the referees' comments, the manuscript has been carefully revised. The answers to the comments by referees are enclosed as below.

(1) Development of Fe-based single-atom nanozyme by incorporating hemoglobin into ZIF-8: The topic of Fe-based single atom nanozyme is not new. It has been shown that bovine hemoglobin can be incorporated into ZIF-8 to develop single atom nanozymes (ACS Appl. Mater. Interfaces 2016, 8, 29052–29061). This nanozyme has been used as peroxidase mimic to oxidize TMB for the colorimetric detection. Many other Fe-based single atom nanozymes with oxidase-like activity have been reported in a review (Nano Res. 2023, 16, 1992–2002). The mechanism of the structure-dependent oxidase-like activity of Fe-N/C nanozyme has also been reported (Chem. Commun. 2019, 55, 5271–5274).

Response:

Thank you for your constructive comments. Developing artificial enzymes with the excellent catalytic performance of natural enzymes has been a long-standing goal for

chemists. Fe single-atom catalysts coordinated with N atoms (Fe-N-C) exhibit well-defined atomic structures and electronic coordination environments and thereby deliver various enzyme-like catalytic activity. Unfortunately, majority of them were synthesized *via* pyrolysis at high temperatures, leading to structural collapse and part of the buried Fe-N_x units inaccessible to biomolecules. Also, the strong stacking of those N-doped graphite carbon in various Fe-N-C nanozymes generally induces the frustrated diffusion of bio-substrates to metal sites. Hence, various methods including spatial confinement, defect/vacancy engineering and coordination modulations have been developed to solve those problems. However, only part of those inadequacies could be overcome. Thus, seeking for a new synthetic strategy of single-atom metal nanozymes to simultaneously achieve the atomic metal dispersion, modulated electronic structure, elevated mass transport and tailorable coordination environment is still on high demands.

In this work, we report a straightforward hemoglobin-mediated approach to efficiently distribute iron atoms homogeneously and in an atomically dispersed fashion across the nitrogen-doped carbon support. Meanwhile, the as-prepared three-coordinated single-atom Fe nanozyme possesses significantly higher specific surface area and mesoporosity to facilitate the mass transport and exposure of active Fe sites, and exhibits ideal oxidase-like catalytic activity. The benefits of our approach could be described as below.

- (1) pFeSAN with the unsymmetrically coordinated Fe-N₃ active sites was synthesized successfully by using Hb as Fe source. This synthesis represented a simple, facile and scalable one.
- (2) Evenly distributed Fe atoms in Hb effectively avoided the agglomeration of active sites during pyrolysis and created mesoporous structure (3~4 nm) in the pFeSAN, thereby maximumly exposing the atomic Fe sites and significantly facilitating the mass transfer of reactants/products during the catalytic process.
- (3) pFeSAN delivered outstanding oxidase-like activity, which was 3.3- and 8791- times higher than those of Fe-N₄ and Fe₃O₄ nanozymes, respectively.
- (4) Mechanism investigations illustrated that pFeSAN underwent a catalytic pathway of the four-electron reduction of O₂ into H₂O, being identical to that of CcO, which was very different from the majority of the previously reported

oxidase-like nanozymes with the generation of reactive oxygen species (ROS).

- (5) pFeSAN as a highly-performed nanozyme exhibited a much higher upper detection limit of GSH at 1 mM, which was 2.5 to 40-fold higher than those of the previously reported investigations (Table R4).
- (6) Visual and rapid detection of tumor tissues through GSH colorimetric analysis was achieved for the first time, which was expected to help the effective resection of tumor tissues.

Herein, we detailed comparisons the previous works listed by the Reviewer and wished to clearly illustrate the value and innovation of our work.

ACS Appl. Mater. Interfaces, 2016, 8, 29052–29061:

In this work, hemeprotein was directly embedded in ZIF-8 and the hemeprotein-embedded ZIF-8 as peroxidase mimic applied for H₂O₂ and phenol detection. The ZIF-8@BHb hybrid composite is a MOF-based nanozyme, not a Fe single atom nanozyme. Compared with our work, the material structure, type of mimic-enzyme activity and detection substrate of the above studies are very difference (Table R1). Thus, the two studies present different research contents.

Table R1. Comparison of pFeSAN and ZIF-8@BHb hybrid composites.

	Catalyst	Mimic function	Application
Our work	Mesoporous Fe-N₃	Oxidase-like (OXD)	GSH detection
ACS Appl. Mater. Interfaces , 2016 , 8, 29052.	ZIF-8@BHb hybrid composite	Peroxidase-like (POD)	H ₂ O ₂ and phenol detection

Nano Res. 2023, 16, 1992–2002:

Despite many Fe-based single atom nanozymes with oxidase-like activity have been reported. Most Fe single atom nanozymes are manufactured by forming an Fe-N_x structure, which is achieved by using inorganic metal precursors and nitrogen-doped carbon support under the high-temperature treatments. **In this process, the drawback is that Fe can aggregate to block the pores and the active site of the Fe-N_x can**

collapse. Moreover, this process generally requires the further acid treatment for the removal of large particles. As catalyst activity decreases of the active center and mass transfer properties, the Fe single atom nanozymes typically exhibit lower performance. To solve these issues, we demonstrated a bioinspired synthetic strategy for the massive preparation of highly active porous three-coordinated Fe single-atom nanozymes (pFeSAN) with oxidase-like activity by using biomineralized hemoglobin-containing ZIF as pyrolytic templates. Hemoglobin is an iron protein that contains nitrogen and metals that are useful as an ideal template and Fe source for bio-inspired Fe single-atom nanozyme. The iron-porphyrin contained in the hemoglobin plays an important role in forming the Fe-N_x active sites during the heat treatment process. Compared to previous studies, pFeSAN with Fe-N₃ coordination and mesoporous structure increased the substrate transfer and thereby exhibited an outstanding oxidase-like activity.

The paper (*Nano Res.* 2023, 16, 1992–2002) is a review article, which summarizes the recent progress of single-atom nanozymes in biomedicine. In this Review article, there is no general strategy to simultaneously achieve atomically dispersed, mesoporous structure and a well-regulated single-atom coordination environment (Table R2). Moreover, the mechanisms of O₂ activation and electrons transfer in the oxidase-like reaction of Fe-based single atom nanozymes are not particularly clear. Therefore, compared to other Fe-based single atom nanozymes, pFeSAN *via* one-step synthesis has the advantages of facile preparation, atomically dispersed Fe single atoms, mesoporous structure, Fe-N₃ coordination and excellent oxidase-like activity, making it a promising nanozyme for GSH detection and other biological applications. To further clarify the reaction mechanism of the pFeSAN, we investigated the O₂ activation and electron-transfer by EPR, DFT and electrochemical analysis. We believe this study will inspire other high-performance Fe-based single atom nanozymes development.

Table R2. Comparison of pFeSAN and other Fe single-atom nanozymes.

Fe source	Mesporous	Template	Aftertreatment	Mimic function	Applications	Ref
Hb	Yes	Hb	None	OXD	GSH detection	Our work
FePc	None	None	HCl	OXD	Antibacterial	Sci. Adv. , 2019, 5, eaav5490.
(NH ₄) ₂ Fe(SO ₄) ₂	Yes	F127	PEG	POD/CAT	Anti-tumor	Biomaterials , 2022, 281, 121325.
Fe(acac) ₃	None	None	H ₂ SO ₄	OXD/POD	Anti-tumor	ACS Nano , 2022, 16, 1, 855
Fe(acac) ₃	None	None	Lyophilization	POD	Antibacterial	Small , 2019, 15, e1901834.
Fe(NO ₃) ₃	None	MgO	HNO ₃	POD	Glucose detection	Small , 2020, 16, e2002343.
Fe(OAc) ₂	None	MgO	HNO ₃	POD	Osteosarcoma treatment	Adv. Mater. , 2021, 33, e2100150.
Fe(NO ₃) ₃	None	SiO ₂	NaOH	POD	Anti-tumor	Adv. Mater. , 2022, 34, e2107088.
FePc	None	None	H ₂ SO ₄	CAT/SOD	Anti-oxidant	Chem. Commun. , 2019, 55, 159.
Fe(NO ₃) ₃	None	None	Lyophilization	POD	Butyrylcholinesterase detection	Biosens. Bioelectron. , 2019, 142, 111495.
FeCl ₂	None	None	Lyophilization	POD	H ₂ O ₂ detection	Anal. Chem. , 2019, 91, 11994.
FeCl ₃	None	SiO ₂	HF	POD	Acetylcholinesterase detection	Small , 2019, 15, e1903108.
Fe(NO ₃) ₃	None	None	None	POD	Acetylcholinesterase detection	ACS Nano , 2022, 16, 2, 2997.
Fe(OAc) ₂	None	None	H ₂ SO ₄	OXD	GSH detection	Chem. Commun. , 2019, 55, 5271.

Chem. Commun. 2019, 55, 5271–5274:

In this work, authors suggest that the $\cdot\text{O}_2^-$ radical is the main active species in the oxidase-like reaction of Fe–N/C–CNTs. However, ROS were undetectable by using either ROS quenchers and electron paramagnetic resonance spectra in our work

(Supplementary Fig. 32). These results confirmed that pFeSAN mediated the complete reduction of O₂ to H₂O without releasing free ROS, indicating that the reaction mechanisms of the structure-dependent oxidase-like activity of Fe–N/C–CNTs and pFeSAN were very different. Meanwhile, the mechanism investigations illustrated that pFeSAN underwent a catalytic pathway of the four-electron reduction of oxygen into H₂O, identical to that of natural *CcO*. We also verified that pFeSAN likely followed an oxygen atom transfer mechanism similar to that of the natural *CcO* by the Fe(IV)=O intermediate, different from majority of previously reported oxidase mimics.

Table R3. Comparison of pFeSAN and Fe–N₃/C.

	Catalyst	Acid treatment	Free ROS
Our work	Mesoporous FeN₃	Not required	None
Chem. Commun. , 2019, 55, 5271.	Fe–N ₃ /C	H ₂ SO ₄ (1 M), 5 h	·O ₂ [–]

Overall, our study is a new synthetic strategy by employing the biomineralized hemoglobin@ZIF-8 as sacrificial templates and Fe sources to prepare the atomically dispersed Fe single-atom nanozyme (featured by Fe–N₃ coordination) within the mesopores of carbon support. Benefiting from the simple and facile one-step Hb-templated strategy, pFeSAN shows **six key advantages (Response Letter: Page R13-R14)** over the conventional FeSAN in terms of catalyst structures, performance of oxidase-like, and GSH detection, as mentioned above as well as in the revised manuscript. We hope that we illustrate the novelty of our investigations as well as distinguish the present study from previous investigations.

(2) Single atom nanozymes for the detection of glutathione (GSH). Nanozymes and Fe–N–C-based single atom nanozymes have been used extensively for the detection of glutathione (*Journal of Materiomics*, 2022, 8, 1251-1259). Mn₃O₄ microspheres have been used as an oxidase mimic for rapid detection of glutathione (*RSC Adv.*, 2019,9, 16509-16514). Light-responsive MOF as an oxidase mimic for cellular GSH detection

(Anal. Chem. 2019, 91, 13, 8170–8175). MnO₂ nanosheets as an artificial enzyme to mimic oxidase for rapid and sensitive detection of glutathione (Biosensors & Bioelectronics, 2017, 90, 69-74). Therefore, the present study lacks novelty in the detection of glutathione.

Response:

Thank you for this comment. Complete surgical resection is the ideal first-line treatment for most malignancies. Compared with biological normal cells, the concentration of GSH in cancer cells is considerably greater, reaching **0.5–1.0 mM**, which is about 1000 times that of normal cells, making it one of the most significant signal molecules to diagnose cancer. Therefore, the goal of complete surgical resection would be facilitated by GSH imaging that enables more precise visualization of tumor margins. However, due to its relatively high concentration, a high-performance detection system for GSH analysis is highly desired to provide a rapid localization of tumor area *via* GSH visualization detection. Unfortunately, the direct detection of GSH in the mM range has been recognized as a difficult challenge. The concentrations of GSH in tumor tissue are usually higher than the upper limit of GSH detection for those previously reported nanozymes (Table R4). Hence, the detection range of these above-mentioned nanozymes is too narrow to detect cell and tissue samples directly, as shown in Table R4. To address this challenge, our work focused on developing a high-performance detection system for the GSH detection using biomineralized hemoglobin@ZIF-8 as sacrificial templates and Fe sources to prepare highly dispersed and FeN₃-coordinated single-atoms nanozyme. Based on these properties, a colorimetric method was developed to detect GSH, which presented a wide linear detection range of **50 nM–1.0 mM** for GSH and successfully employed as a colorimetric probe for GSH visualization in tumor tissues.

Table R4. Comparison of the pFeSAN with other nanozymes for GSH detection.

Materials	Linear range (μM)	LOD (μM)	Reference
pFeSAN	0.05-1000	0.0024	This work
Fe-N-C SANs	100-400	78.3	J. Materiomics , 2022, 8, 1251.
MnO ₂	1-25	0.3	Biosens. Bioelectron. , 2017, 90, 69.
PSMOF	0-40	0.68	Anal. Chem. , 2019, 91, 8170.
Mn ₃ O ₄	50-60	0.889	RSC Adv. , 2019, 9, 16509.

In our work, the exceptionally high detection upper limit of pFeSAN is 2.5 to 40 times higher than that of these conventional nanozymes, making it suitable for detecting GSH with high levels (Table R4). The excellent detection performance of GSH by the pFeSAN originates from its synergistic advantages comprising highly dispersed metal single-atoms, the presence of mesopores, and well-regulated coordination environments of the single-atoms. We further utilize pFeSAN-DAB system as a GSH sensor to realize in vitro quantitative GSH visualization for Hep 1-6 cells and tumor tissue with high GSH states have been accurately distinguished by visualization detection.

Besides, the comparison illustrated the dramatical difference of the pFeSAN from the previously reported nanozymes, which could be attributed to the unique structural features of pFeSAN and an enzyme-like catalytic pathway of the four-electron O₂-to-H₂O reduction. *The outstanding performance, unique structures and the natural enzyme-like catalytic pathway of pFeSAN indeed reflect the novelty of the dedicatedly designed catalysts for GSH detection, which serve as a real-time, facile, rapid (~6 min) and precise visualization analysis methodology of tumors and shows its potential for diagnostic and clinic applications.*

(3) The concept of using GSH as biomarker for visualization of cancer cells is not entirely new. There are many recent reports in the literature which highlight the concept. For example, a MOF has been reported to exhibit oxidase-like activity by oxidizing TMB to oxTMB, which has been used as a colorimetric probe for GSH detection. The oxidase mimic has been used to analyze the GSH level in the lysates of normal and cancer cells (*Anal. Chem.* 2019, *91*, 8170–8175). There are many other reports which describe the use of nanozymes for tumor visualization through GSH detection.

Response:

Thank you very much for pointing this out. The author developed a light-dependent metal–organic framework with oxidase-like activity and colorimetric detection of cancer cells through GSH detection after cell lysis treatment (Ref. 53: *Anal. Chem.* 2019, *91*, 8170–8175). In our work, pFeSAN was applied to monitor the GSH levels in normal and cancer cells **without additional requirements of light irradiation and complex pretreatment of cells**. Importantly, methods for the visualization detection of tumor tissue through GSH has not been reported yet, which brought the possibility to realize specific detection in practical applications. The visualized analysis of the GSH in tumor tissue is gaining interests to improve surgical safety and to promote surgical therapeutic effects.

Table R5. Comparison of the pFeSAN with PSMOF for GSH detection.

	Catalyst	Light dependent	Cell pretreatment	Intratumoral GSH detection
Our work	pFeSAN	Not required	Without pretreatment	Yes
Anal. Chem. 2019, 91 , 8170.	PSMOF	300 W Xe lamp	Cell lysis	None

Reviewers' Comments:

Reviewer #1:

Remarks to the Author:

This paper presents an innovative biomimetic synthetic strategy for the synthesis of porous Fe-N3 single atom nanozymes (pFeSAN) using natural enzyme as a template, and the nanozyme exhibits highly efficient oxidase-like activity and potential applications in diagnostics. This research has value for the researchers in the related areas. A thorough, point-by-point response to each point raised by reviewers has been made. So I think this work can be accepted for publication in Nature Communications.

Reviewer #2:

Remarks to the Author:

Authors have improved the manuscript according to the comments, and thus I recommend it publication.

Reviewer #4:

Remarks to the Author:

[Note from the editor: Reviewer #4 was invited to assess the response given to Reviewer #3]

In the manuscript "Bioinspired porous three-coordinated single-atom Fe nanozyme with oxidase-like activity for tumor visual identification via glutathione" rebuttal letter, the authors responded to the three questions raised by the reviewer 3 with lots of discussion. However, it might still not fully address the concerns about the novelty mentioned there.

Author response: (2) Evenly distributed Fe atoms in Hb effectively avoided the agglomeration of active sites during pyrolysis and created mesoporous structure (3~4 nm) in the pFeSAN, thereby maximumly exposing the atomic Fe sites and significantly facilitating the mass transfer of reactants / products during the catalytic process.

Comments: The increase in the transfer of reactants / products usually related to Km value of the enzyme. However, in the manuscript, the authors showed that the pFeSAN showed a really small increase in binding affinity compared to the Fe-N4 they mentioned. This might need further clarification or more thoughts.

Author response: (3) pFeSAN delivered outstanding oxidase-like activity, which was 3.3- and 8791- times higher than those of Fe-N4 and Fe3O4 nanozymes, respectively.

Comments: The oxidase-like activity of the nanozymes were compared using activity ratio which is a not standardized item. How that is calculated and how to use that value to cross-compare with other oxidase-like nanozyme require further clarifications.

Author response: (4) Mechanism investigations illustrated that pFeSAN underwent a catalytic pathway of the four-electron reduction of O2 into H2O, being identical to that of CcO, which was very different from the majority of the previously reported oxidase-like nanozymes with the generation of reactive oxygen species (ROS).

Comments: This finding is not novel. As the nanozymes prepared by the authors only showed OXD-like activity, it is expected that there will be no ROS generated. (Similar to Sci. Adv., 2019, 5, eaav5490). If the authors stated it is a similar catalytic pathway to CcO, is there any of the reaction intermediates or transient species the authors trapped could be compared to CcO?

Author response: (5) pFeSAN as a highly-performed nanozyme exhibited a much higher upper detection limit of GSH at 1 mM, which was 2.5 to 40-fold higher than those of the previously reported investigations (Table R4).

Comments: Previously, other systems based on nanozyme activity detected similar or even higher range. (New J. Chem., 2022,46, 10682-10689)

Point-by-point Response to Reviewers' Comments

Reviewer #1 (Remarks to the Author):

This paper presents an innovative biomimetic synthetic strategy for the synthesis of porous Fe-N₃ single atom nanozymes (pFeSAN) using natural enzyme as a template, and the nanozyme exhibits highly efficient oxidase-like activity and potential applications in diagnostics. This research has value for the researchers in the related areas. A thorough, point-by-point response to each point raised by reviewers has been made. So I think this work can be accepted for publication in Nature Communications.

Response:

We appreciate the Reviewer's positive comment and valuable suggestions, which helped us improve our manuscript.

Reviewer #2 (Remarks to the Author):

Authors have improved the manuscript according to the comments, and thus I recommend it publication.

Response:

We appreciate the Reviewer's positive comment and valuable suggestions, which helped us improve our manuscript.

Reviewer #4 (Remarks to the Author):

[Note from the editor: Reviewer #4 was invited to assess the response given to Reviewer #3]

In the manuscript “Bioinspired porous three-coordinated single-atom Fe nanozyme with oxidase-like activity for tumor visual identification via glutathione” rebuttal letter, the authors responded to the three questions raised by the reviewer 3 with lots of discussion. However, it might still not fully address the concerns about the novelty mentioned there.

We really appreciate the Reviewer’s useful comments and suggestions. We have carefully revised the manuscript based on his/her comments.

Author response: (2) Evenly distributed Fe atoms in Hb effectively avoided the agglomeration of active sites during pyrolysis and created mesoporous structure (3~4 nm) in the pFeSAN, thereby maximumly exposing the atomic Fe sites and significantly facilitating the mass transfer of reactants / products during the catalytic process.

Comments: The increase in the transfer of reactants / products usually related to K_m value of the enzyme. However, in the manuscript, the authors showed that the pFeSAN showed a really small increase in binding affinity compared to the Fe-N₄ they mentioned. This might need further clarification or more thoughts.

Response:

Thanks for the Reviewer’s kind suggestions, which are valuable for improving the accuracy of the manuscript.

For analyzing the catalytic mechanism and acquiring kinetic parameters, the oxidase-like activities of pFeSAN and Fe-N₄ under the same conditions were studied by enzyme kinetics theory and methods. With the Lineweaver-Burk equation, the important

enzyme kinetic parameters such as Michaelis–Menten constant (K_m), maximal velocity (V_{max}), catalytic constant (K_{cat}) and K_{cat}/K_m were presented in Table R1 (*Adv. Mater.*, 2022, 34, e2201736; *Nat. Protoc.*, 2018, 13, 1506-1520.). K_m was identified as an indicator of enzyme affinity to substrates. Smaller K_m values thus indicate a stronger affinity between the enzyme and the substrate. V_{max} represents the reaction rate when the enzyme is saturated with substrate, and a higher V_{max} value indicates a quicker reaction rate. The K_{cat} value gave a direct measure of the enzymatic catalytic activity. Generally, the maximal velocity of reaction V_{max} and K_{cat} reveals the catalytic activity of enzyme. Increasing the transfer of chemicals not only increase the decrease of K_m but also increase the diffusion of chemicals, leading to the overall enhanced catalytic kinetics. Besides, K_{cat}/K_m is known to be a descriptor of enzyme efficiency and a better indicator to compare two enzymes. The higher catalytic efficiency of pFeSAN signifies more substrate-to-product conversion, which is happening due to its large affinity (low K_m) for TMB and greater proportion of bound substrate conversion to product before its dissociation (large turnover K_{cat}).

Table R1. Comparison of kinetics for pFeSAN and Fe-N₄.

Catalyst	K_m (mM)	V_{max} ($\mu\text{M s}^{-1}$)	K_{cat} (s^{-1})	K_{cat}/K_m ($\text{mM}^{-1}\text{s}^{-1}$)
pFeSAN	0.17	1.67	2.6×10^6	1.53×10^7
Fe-N ₄	0.29	0.036	9.6×10^4	3.31×10^5

From Table R1, the K_m value for pFeSAN (0.17 mM) to the TMB was about 58% of that for Fe-N₄ (0.29 mM), indicating that it has higher affinity with TMB and lower concentration of TMB required to reach the maximal activity of V_{max} . Hence, the V_{max} and K_{cat} values of pFeSAN for TMB showed 46.4-fold and 27.1-fold increases relative to Fe-N₄, verifying the mesoporous structure in improving the oxidase-like performance. Meanwhile, the catalytic efficiency (K_{cat}/K_m) of pFeSAN ($1.53 \times 10^7 \text{ mM}^{-1}\text{s}^{-1}$) is 46.2-fold higher than that of Fe-N₄ ($3.31 \times 10^5 \text{ mM}^{-1}\text{s}^{-1}$). Overall, all these kinetic parameters including Michaelis-Menten constant (K_m), maximal reaction velocity

(V_{\max}), catalytic rate constant (K_{cat}), and K_{cat}/K_m of pFeSAN showed the optimum values than Fe-N₄, indicating a distinct positive contribution of the mesoporous structure (3~4 nm) and higher surface area to the oxidase-like activity of the pFeSAN (Figure R1). The larger pore size can make a great contribution to fast mass transfer.

Figure R1. a. Pore size distribution curves of Fe-N₄ and pFeSAN b. BET surface areas of Fe-N₄ and pFeSAN.

In conclusion, we demonstrated a biomimetic synthetic strategy for scalable synthesis of porous Fe single-atom nanozymes using hemoglobin as both template and Fe-source, which delivered a high oxidase-like activity. The mesoporous features of pFeSAN significantly promoted mass transport and maximumly exposed active iron sites during reaction, which could greatly enhance the oxidase-like activity of pFeSAN. The relative information has been updated in the revised manuscript (Page 16, Line 19–22; Page 17, Line 1–5 and Supplementary Table 2).

Author response: (3) pFeSAN delivered outstanding oxidase-like activity, which was 3.3- and 8791- times higher than those of Fe-N₄ and Fe₃O₄ nanozymes, respectively.

Comments: The oxidase-like activity of the nanozymes were compared using activity ratio which is a not standardized item. How that is calculated and how to use that value to cross-compare with other oxidase-like nanozyme require further clarifications.

Response:

We would like to express our point for your valuable suggestions regarding our work.

In response, we want to clarify that we initially utilized a generic method to calculate the activity of nanozymes in the main text. We recognized the importance of consistency in comparing the activity of different nanozymes.

We calculated the oxidase-like activities of pFeSAN, Fe-N₄ and Fe₃O₄ according to the standardized assay protocol (*Nat. Catal.* 2021, 4, 407-417; *Nat. Protoc.* 2018, 13, 1506-1520). By quantitatively determined the specific activity values (U/mg) of pFeSAN, Fe-N₄ and Fe₃O₄ by measuring the absorption intensity of the nanozyme-catalyzed TMB colorimetric reactions (Figure R2). The specific activity of pFeSAN was determined to be 593 U/mg, which is 3.5 times higher than that of Fe-N₄ (169 U/mg) and 8471 times higher than that of Fe₃O₄ (0.07 U/mg). The high specific activity of pFeSAN following factors: Firstly, the mesoporous structure (3~4 nm) of pFeSAN exhibits larger surface area (705.8 m²/g) than that of Fe-N₄ (561.6 m²/g), and the large specific surface area can help fast mass transfer. Secondly, the pFeSAN exposes more Fe active sites, greatly enhance its oxidase-like activity.

Figure R2. Reaction-time curves of the TMB colorimetric reaction catalyzed by pFeSAN, Fe-N₄ and Fe₃O₄.

The calculation part of the specific activity is as follows:

Calculate the nanozyme activity (units) using the following equation:

$$b_{\text{nanozyme}} = V/(\epsilon \times l) \times (\Delta A / \Delta t)$$

where b_{nanozyme} is the catalytic activity of nanozyme expressed in units. One unit is

defined as the amount of nanozyme that catalytically produces 1 μmol of product per min at room temperature; V is the total volume of reaction solution (μL); ϵ is the molar absorption coefficient of the colorimetric substrate, which is maximized at $39,000 \text{ M}^{-1} \text{ cm}^{-1}$ at 652 nm for TMB; l is the path length of light traveling in the cuvette (cm); A is the absorbance after subtraction of the blank value; and $\Delta A/\Delta t$ is the initial rate of change in absorbance at 652 nm min^{-1} .

Calculate the specific activity of the nanozyme (U mg^{-1}) by

$$a_{\text{nanozyme}} = b_{\text{nanozyme}}/[m]$$

where a_{nanozyme} is the specific activity expressed in units per milligram (U mg^{-1}) nanozymes, and $[m]$ is the nanozyme weight (mg) of each assay.

In the revised manuscript, we have updated these data according to the Reviewer's suggestion (Page 14, Fig. 4f; Page 16, Line 1-7; Page 33, line 3-16).

Author response: (4) Mechanism investigations illustrated that pFeSAN underwent a catalytic pathway of the four-electron reduction of O_2 into H_2O , being identical to that of CcO, which was very different from the majority of the previously reported oxidase-like nanozymes with the generation of reactive oxygen species (ROS).

Comments: This finding is not novel. As the nanozymes prepared by the authors only showed OXD-like activity, it is expected that there will be no ROS generated. (Similar to *Sci. Adv.*, 2019, 5, eaav5490). If the authors stated it is a similar catalytic pathway to CcO, is there any of the reaction intermediates or transient species the authors trapped could be compared to CcO?

Response:

Thank you for your valuable comments. Previous studies demonstrated that the reactive intermediate of Fe(IV)=O was very important for the catalytic oxidative reactions of natural CcO (*Chem. Rev.* 2018, 118, 2491–2553). As the common oxidant, the intermediate of Fe(IV)=O , which usually presents in the catalytic cycle of natural

oxidases, is considered as the active transient state. To verify the presence of the Fe(IV)=O intermediate of O₂ activation process by pFeSAN, the electron paramagnetic resonance spectrum of the pFeSAN-enabled oxidation with excessive phenyloxiodine was recorded at 77K (Figure R3). A typical diamond-shaped sign signal at g=2.03, consistent with η^2 -peroxo heme species, indicated the formation of Fe(IV)=O intermediate in pFeSAN for oxidation (Fig. 5f, *Sci. Adv.*, 2019, 5, eaav5490). Therefore, the oxidase-like activity of pFeSAN proceeds through the O₂-to-H₂O pathway, and similar to the reaction process with CcO.

Figure R3. EPR spectra of pFeSAN in the presence of phenyloxiodine at 77 K.

Previous study (*Sci. Adv.*, 2019, 5, eaav5490) demonstrated that the FeN₅ SA/CNF showed oxidase-like activity and no generation of ROS. However, the experimental verification was lacking in that study. In our work, to further explore the electron transfer path during the oxidation, the rotating ring-disk electrode (RRDE) tests were performed. Figure R4 showed that the H₂O₂ yield of pFeSAN remained below 7.5% over a wide potential range of 0.1–0.8 V. Derived from the RRDE test, the average electron transfer number (n) of pFeSAN was 3.7, indicating the oxygen activation on the pFeSAN through a four-electron oxygen reduction reaction pathway (*Nano Energy*, 2021, 83, 105798). This process requires four H⁺ and four electrons (O₂+4H⁺+4e⁻→2H₂O) for the complete O₂-to-H₂O reduction. Thus, the electrochemical understandings of these stepwise proton and electron transfers reveal the essence of pFeSAN for its oxidase-like performance. In our work, the exploration of oxidase-like

reaction by the RRDE tests would help us to understand the mechanism of oxidase-like reactions.

Figure R4. Calculated electron transfer number derived from rotating ring-disk electrode and H₂O₂ yields of the pFeSAN.

In the revised manuscript, we have highlighted the related information in the main context (Page 20, Line 1–9).

Author response: (5) pFeSAN as a highly-performed nanozyme exhibited a much higher upper detection limit of GSH at 1 mM, which was 2.5 to 40-fold higher than those of the previously reported investigations (Table R4).

Comments: Previously, other systems based on nanozyme activity detected similar or even higher range. (New J. Chem., 2022, 46, 10682-10689)

Response:

Thank you for your evaluation. In this GSH detection system (*New J. Chem.*, 2022, 46, 10682-10689), a novel nanozyme based on the ultrathin two-dimensional metal-organic framework nanomaterial D-ZIF-67 was prepared and characterized. D-ZIF-67 exhibited significant oxidase-like activity due to the large specific surface area of the two-dimensional sheet structure as well as the large number of active sites exposed compared to crystalline MOFs. By taking advantage of the excellent property of D-ZIF-67, authors constructed an effective and sensitive colorimetric sensor for visual GSH

detection, the detected GSH concentrations ranged between 0.5–10 μM .

In our work, pFeSAN provided a wider detection range of 50 nM to 1 mM, higher than those previous literatures (Table R2). The excellent detection performance of GSH by the pFeSAN originates from its synergistic advantages comprising highly dispersed metal single-atoms, the presence of mesopores, and well-regulated coordination environments of the single-atoms. We further utilize pFeSAN-DAB system as a GSH sensor to realize in vitro quantitative GSH visualization for Hep 1-6 cells and tumor tissue with high GSH states have been accurately distinguished by visualization detection. Notably, methods for the visualization detection of tumor tissue through GSH has not been reported yet, which brought the possibility to realize specific detection in practical applications. The visualized analysis of the GSH in tumor tissue is gaining interests to improve surgical safety and to promote surgical therapeutic effects. We have checked the literature carefully and added this literature in the revised manuscript to support this work (Ref. 57: *New J. Chem.*, 46, 10682–10689 (2022)).

Table R2. Comparison of our approach with other colorimetric detection systems of GSH.

Materials	Linear range	LOD
pFeSAN	0.05-1000 μM	0.0024 μM
AuNPs	1-40 μM	0.013 μM
Au nanoclusters	2-25 μM	0.42 μM
PSMOF	1-20 μM	0.68 μM
Acr ⁺ -Mes	0.1-40 μM	0.1 μM
MnO ₂	0.3-15 μM	0.11 μM
TiO ₂ /MoS ₂	0.05-1 μM	0.05 μM
Fe-N-C SANs	100-400 μM	78.3 μM

Reviewers' Comments:

Reviewer #4:

Remarks to the Author:

The authors have addressed the comments. There is no further comment.

Point-by-point Response to Reviewers' Comments

Reviewer #4 (Remarks to the Author):

The authors have addressed the comments. There is no further comment.

Response:

We appreciate the Reviewer's positive comment and valuable suggestions, which helped us improve our manuscript.